# Pharmacological induction of autophagy reduces inflammation in macrophages by degrading immunoproteasome subunits

Jiao Zhou[1,2☯], Chunxia Li[1☯], Meng Lu[1], Gaoyue Jiang[1], Shanze Chen[3], Huihui Li[4], Kefeng Lu[1]*

**1** Department of Neurosurgery, State Key Laboratory of Biotherapy, West China Hospital, Sichuan University and the Research Units of West China, Chinese Academy of Medical Sciences, Chengdu, China, **2** National Clinical Research Center for Geriatrics, West China Hospital, Sichuan University and Collaborative Innovation Center of Biotherapy, Chengdu, China, **3** Department of Respiratory and Critical Care Medicine, First Affiliated Hospital of Southern University of Science and Technology, Second Clinical Medical College of Jinan University, Shenzhen People's Hospital, Shenzhen Institute of Respiratory Diseases, Shenzhen, China, **4** West China Second University Hospital, Sichuan University, Chengdu, China

☯ These authors contributed equally to this work.
* lukf@scu.edu.cn

**Data Availability Statement:** All relevant data are within the paper and its Supporting Information files.

## Abstract

Defective autophagy is linked to proinflammatory diseases. However, the mechanisms by which autophagy limits inflammation remain elusive. Here, we found that the pan-FGFR inhibitor LY2874455 efficiently activated autophagy and suppressed expression of proinflammatory factors in macrophages stimulated by lipopolysaccharide (LPS). Multiplex proteomic profiling identified the immunoproteasome, which is a specific isoform of the 20s constitutive proteasome, as a substrate that is degraded by selective autophagy. SQSTM1/p62 was found to be a selective autophagy-related receptor that mediated this degradation. Autophagy deficiency or p62 knockdown blocked the effects of LY2874455, leading to the accumulation of immunoproteasomes and increases in inflammatory reactions. Expression of proinflammatory factors in autophagy-deficient macrophages could be reversed by immunoproteasome inhibitors, confirming the pivotal role of immunoproteasome turnover in the autophagy-mediated suppression on the expression of proinflammatory factors. In mice, LY2874455 protected against LPS-induced acute lung injury and dextran sulfate sodium (DSS)-induced colitis and caused low levels of proinflammatory cytokines and immunoproteasomes. These findings suggested that selective autophagy of the immunoproteasome was a key regulator of signaling via the innate immune system.

## Introduction

Inflammation is an essential immune response that enables survival during pathogenic microorganism infection or tissue injury [1,2]. Various extrinsic and intrinsic inflammatory stimuli include infectious microorganisms (such as viruses and bacteria), cancer cells, or dead cells [3–6]. Although inflammation is a vital component of a healthy immune response in the host,

**Funding:** This work was supported by the National Key R&D Program of China under grant 2017YFA0506300 (to K.L.) and the National Natural Science Foundation under grant 32022020 (to K.L.) and 81902997 (to H.L.). The funders had no role in study design, data collection and analysis, decision to publish, or preparation of the manuscript.

**Competing interests:** The authors have declared that no competing interests exist.

**Abbreviations:** BMDM, bone marrow-derived macrophage; COVID-19, Coronavirus Disease 2019; DSS, dextran sulfate sodium; gRNA, guide RNA; HE, hematoxylineosin; IBD, inflammatory bowel disease; ICU, intensive care unit; iNOS, inducible nitric oxide synthase; LMP2, low molecular mass polypeptide; LPS, lipopolysaccharide; NO, nitric oxide; PAMP, pathogen-associated molecular pattern; PASEF, parallel accumulation serial fragmentatio; PBS, phosphate-buffered saline; PM, peritoneal macrophage; qRT-PCR, quantitative real-time polymerase chain reaction; ROS, reactive oxygen species; SARS-CoV-2, Severe Acute Respiratory Syndrome Coronavirus 2; TNF-α, tumor necrosis factor-α.

rampant inflammation results in various pathologies such as sepsis, rheumatoid arthritis, acute lung injury, chronic respiratory diseases, inflammatory bowel disease (IBD), cancer, aging, and even death [7–16]. One very recent example of inflammation-related disease is Coronavirus Disease 2019 (COVID-19), which is caused by Severe Acute Respiratory Syndrome Coronavirus 2 (SARS-CoV-2). Serious COVID-19 is associated with inflammation, and there is an emphasis on cytokine storm syndrome [17–25]. Uncontrolled macrophage and monocyte activation in response to SARS-CoV-2 infection causes a rampant subsequent inflammatory response that result in acute respiratory distress syndrome and end-organ injury, including lung fibrosis [20,26,27]. Anti-inflammatory factor therapies have been used to inhibit the development of cytokine storms in COVID-19 patients [28–32]. In addition to viruses, pathogenic bacterial infection is also a robust cause of inflammation and death. Approximately 50% of patients in intensive care units (ICUs) develop severe sepsis caused by bacterial infection, and a 29% mortality rate has been shown in sepsis patients [33–35]. Mortality is mostly attributed to the cytotoxic effects of lipopolysaccharide (LPS) (endotoxin), a component of the outer membrane of gram-negative bacteria [36–39]. Various and large numbers of inflammatory factors, such as tumor necrosis factor-α (TNF-α), interleukin-1β (IL-1β), and interleukin-6 (IL-6), and high levels of intracellular reactive oxygen species (ROS), are substantially induced in LPS-stimulated macrophages both ex vivo and in vitro [40–44].

The immunoproteasome is an important mediator of inflammation in immune cells such as macrophages and nonimmune cells, and its expression is triggered by inflammatory signals [45–57]. The immunoproteasome is a specific isoform of the 20s constitutive proteasome that is quickly formed (in about 20 min) by replacing the β1, β2, and β5 subunits of the 20s proteasome with different counterparts known as β1i (low molecular mass polypeptide 2, LMP2), β2i (multi-catalytic endopeptidase complexlike-1, MECL-1, also known as LMP10), and β5i (low molecular mass polypeptide 2, LMP7), while mixed proteasomes consisting of both, classical and immunoproteasome subunits, can be formed [45,46,50,58]. Originally, the immunoproteasome was found to hydrolyse proteins into peptides loaded into the MHC-I complex as antigens [47,59,60]. Apart from antigen processing, the immunoproteasome has been found to promote macrophage polarization [61], brain inflammation [62–64], diabetic nephropathy [65], the production of inflammatory cytokines [66–70], myometrium inflammation associated with preterm labor [71], diet-induced atherosclerosis [72], various inflammatory and autoimmune diseases [66,68,73–78], colitis and colitis-associated cancers [79–82], viral infection [83–90], the activation of lymphocytes [91,92], and cardiac hypertrophy [93,94]. The necessity of reducing excessive immunoproteasomes has been highlighted, and immunoproteasome inhibitors have become promising drug candidates for various diseases, such as hematologic malignancies, autoimmune, and inflammatory diseases [95–102]. However, it is unknown whether and how degradation of the immunoproteasome is achieved.

Macroautophagy (abbreviated as autophagy) is a catabolic pathway for bulk transport through double-layered membrane vesicles (autophagosomes) and the eventual degradation of intracellular components in lysosomes [103–106]. The autophagy pathway is highly conserved among eukaryotic organisms and plays a fundamental role in various physiological and pathological conditions [107–109]. The autophagy pathway is a dynamic process that functions to maintain cellular homeostasis through the selective and bulk degradation of cytoplastic cargoes such as toxic protein aggregates, unnecessary or damaged organelles, and intracellular pathogens [110–113]. Autophagy dysfunction has been linked to several proinflammatory diseases [114–118], such as rheumatoid arthritis, lupus, and IBD [119–122]. Variations in the human autophagy genes ATG16L1, NOD2, IRGM, ATG5, and ATG7 is associated with chronic inflammatory diseases including Crohn's disease [123–127].

Thus, it is important to clarify whether and how autophagy targets the immunoproteasome for degradation and the involvement of this degradation in limiting inflammation. In the present study, we identified the pan-FGFR inhibitor LY2874455 as an effective anti-inflammatory treatment acting through the activation of autophagy. We utilized LPS-induced inflammatory responses in murine macrophages as a functional screen and identified the pan-FGFR inhibitor LY2874455 as an effective inhibitor of inflammation. We found that the stimulation of autophagy by LY2874455 promoted the autophagic degradation of the immunoproteasomes in LPS-induced macrophages. LY2874455 inhibited the AKT-mTOR axis to activate autophagy, and autophagic degradation of the immunoproteasome was mediated by the receptor SQSTM1/p62. LY2874455-activated autophagy dramatically inhibited the production of inflammatory cytokines, and blocking autophagy with inhibitors, ATG7 knockout, or knockdown of the receptor p62 reversed this inhibition. The protective role of LY2874455-induced autophagic degradation of the immunoproteasome was verified by in vivo mouse models with acute lung injury induced by LPS or with IBD induced by dextran sulfate sodium (DSS). These findings suggested that selective autophagic degradation of the immunoproteasome was a pivotal regulator of innate inflammatory signaling.

## Results

### The pan-FGFR inhibitor LY2874455 effectively suppressed expression of proinflammatory factors in LPS-stimulated macrophages

Because numerous diseases are caused by rampant inflammation, we screened for chemicals with potential suppressive effects on inflammation in the LPS-treated macrophage cell line RAW264.7. Nitrous oxide (NO) levels were measured because NO is generated at increased levels during inflammation [128]. RAW264.7 cells were cultured in 96-well plates and treated with LPS and in total 5,618 compounds including the FDA-approved drug library (MedChem-Express, Cat. HY-L022, 2669 compounds), PI3K/Akt/mTOR compound library (MedChem-Express, Cat. HY-L015, 456 compounds), and kinase inhibitor library (MedChemExpress, Cat. HY-L009, 2493 compounds). The NO concentration in the culture supernatant was measured in a high-throughput manner with 3 replicate wells for each compound. We found that LY2874455 was the most effective anti-inflammatory chemical by measuring NO levels (only 27 chemicals, including LY2874455, are shown and several other chemicals showed less effects) (S1A and S1B Fig). LY2874455 ((R)-(E)-2-(4-(2-(5-(1-(3,5-dichloropyridin-4-yl) ethoxy)-1H-indazol-3yl) vinyl)-1H-pyrazol-1-yl) ethanol) (Fig 1A) is a novel pan-FGFR inhibitor that shows good tolerability and activity in solid organ cancer patients [129,130]. In addition to suppressing NO generation in LPS-stimulated RAW264.7 macrophages without affecting cell viabilities (Figs 1B, S1C and S1D), LY2874455 also reduced the levels of ROS (shown by DCFH-DA staining) in LPS-stimulated RAW264.7 cells (Fig 1C and 1D). The levels of proinflammatory cytokines, such as tumor necrosis factor-α (TNF-α) and interleukin-6 (IL-6), and inducible nitric oxide synthase (iNOS) were dramatically induced by LPS in RAW264.7 macrophages, and LY2874455 completely reversed this effect (Figs 1E–1H and S1E–S1H). Suppression on the expression of proinflammatory factors by LY2874455 was repeatedly observed in peritoneal macrophages (PMs) (S1I–S1M Fig) and bone marrow-derived macrophages (BMDMs) (S1N–S1R Fig) obtained from mice [131,132] and in human THP-1 monocytes (S1S and S1T Fig). Beside the mRNA levels, the protein levels of inflammatory factors induced by LPS treatment were also reduced by LY2874455 treatment (S1U and S1V Fig). NF-κB signaling is a classic proinflammatory regulator that mediates the induction of various inflammatory cytokines [133–136]. The IκB kinase (IKK) complex, which contains IKK-α, IKK-β, and IKK-γ, activates the inflammatory transcription factor NF-κB. IKK-β phosphorylates

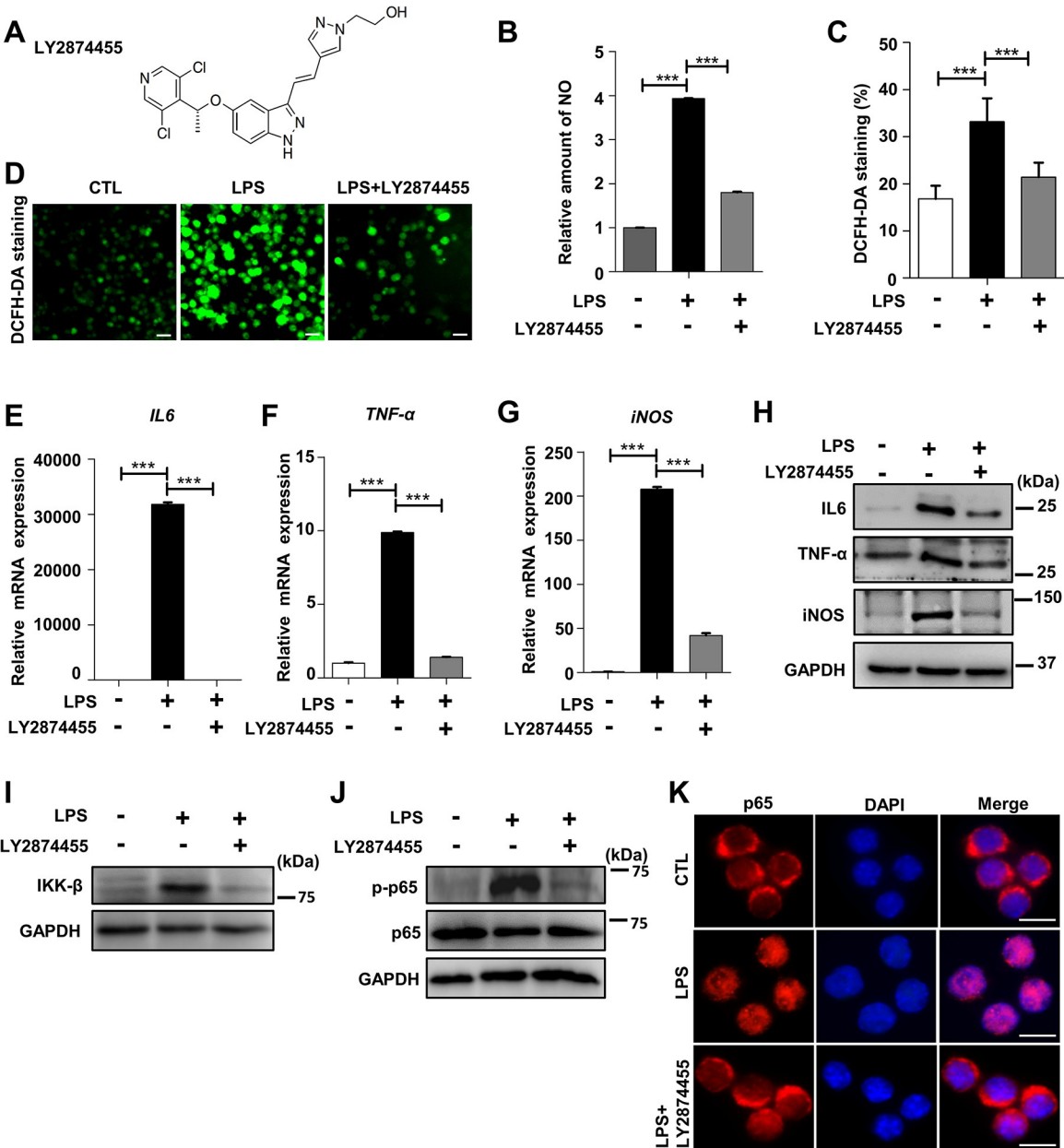

**Fig 1. LY2874455 attenuated the inflammatory reactions in LPS-stimulated macrophages. (A)** The chemical structure of LY2874455. **(B–K)** Macrophage RAW264.7 cells were stimulated with LPS (20 ng/ml) for 24 h together with or without LY2874455 (2 μm). **(B)** LY2874455 abolished the release of NO from LPS-stimulated RAW264.7 cells. NO concentration in culture supernatant was measured and shown as fold change compared to the control group. Similar results were obtained from 3 independent replicated experiments. **P < 0.01, ***P < 0.001. **(C)** LY2874455 abolished the generation of ROS in LPS-stimulated RAW264.7 cells. After treatment with LPS or/and LY2874455, RAW264.7 cells were washed and labeled with DCFH-DA, a ROS probe. Quantitative measurement of cellular DCFH-DA staining was performed by flow cytometry. Similar results were obtained from 3 independent experiments. **(D)** A set of representative images showing fluorescent staining of ROS by DCFH-DA probe were shown. Scale bars: 10 μm. **(E–G)** LY2874455 suppressed the expression of proinflammatory cytokines and NO synthetase in macrophages stimulated with LPS. The expression of proinflammatory cytokine genes (IL-6, TNF-α) and NO synthetase (iNOS) was checked by qRT-PCR. **(H–J)** Total proteins were extracted from RAW264.7 cells and used for western blot analysis of protein levels of IL6, TNF-α, iNOS, IKK-β, p65, and phosphorylated p65 (p-p65). For the similar experiments shown by immunoblots, 3 independent experiments were conducted and representative images were shown. The blots were quantified with densitometric values and statistical significance was analyzed (S1W–S1AE Fig). **(K)** LY2874455 suppressed the nucleus translocation of phosphorylated p65 in macrophages stimulated with LPS. RAW264.7 cells were stimulated with LPS (20 ng/ml) for 2 h with or without LY2874455 (2 μm). Representative images of double immunostaining for p65 and nucleus (DAPI) were shown. Scale bars: 10 μm. *: P < 0.05, **: P < 0.01, ***: P < 0.001, ****: P < 0.0001, NS: no statistical difference. The data underlying the graphs shown in the figure can be found in S1 Data. iNOS, inducible nitric oxide synthase; LPS, lipopolysaccharide; NO, nitric oxide; qRT-PCR, quantitative real-time polymerase chain reaction; ROS, reactive oxygen species.

inhibitory IκBα to promote its polyubiquitination and proteasomal degradation, and then the classic p65/p50 heterodimer is freed to translocate to the nucleus and mediate the transcription of NF-κB target genes, including various cytokines. The LPS-induced protein levels of IKK-β in RAW264.7 cells were reduced by LY2874455 (**Fig 1I**). The activated form of p65 (phosphorylated) and its translocation into the nucleus were also reversed by LY2874455 in LPS-stimulated RAW264.7 cells (**Fig 1J and 1K**).

Taken together, these results demonstrated that the pan-FGFR inhibitor LY2874455 effectively suppressed the inflammatory responses in LPS-stimulated macrophages.

## LY2874455 suppressed lung and intestinal inflammation in mouse models

After demonstrating the suppressive effect of LY2874455 on production of proinflammatory factors at the cellular level, we then analyzed whether LY2874455 could affect inflammatory mouse models with acute lung injury and septicopyemia induced by LPS [137,138] or with IBD induced by DSS. Histological examination of the lungs of mice stimulated with LPS revealed considerable inflammatory cell infiltration, which is a typical symptom of lung inflammation (**Fig 2A**). Notably, inflammatory cell infiltration was reduced by LY2874455 (**Fig 2A**). The LPS-induced high levels of the proinflammatory cytokine IL-6 in the lung tissues of mice were also significantly reduced by LY2874455 (**Fig 2B**). In addition, survival analysis of LPS-stimulated mice with or without LY2874455 treatment was measured. The survival rate was 100% in the control group but dropped notably in the LPS-treated group, while the survival rate was dramatically increased in mice treated with LY2874455 (**Fig 2C**). These results indicated that LY2874455 suppressed inflammation in mice with acute lung injury and septic pyemia induced by LPS and promoted survival in LPS-treated mice.

Furthermore, DSS-induced colitis model mice were used to analyze the potential gastrointestinal protective effects of LY2874455 (**Fig 2D and 2E**). The results showed that LY2874455 notably ameliorated the reduction in colon length in DSS-treated mice (**Fig 2F**). To confirm the effect of LY2874455 on bowel inflammation and tissue injury, colonic sections of mice were subjected to histological examination. DSS caused severe damage to the mucosa of intestinal villi, and LY2874455 treatment efficiently reduced tissue damage (**Fig 2G and 2H**). The expression levels of the proinflammatory cytokines were reduced in colonic tissues from LY2874455-treated mice (**Fig 2I–2L**), which indicated the suppressive effect of LY2874455 on inflammation in colitis model mice. The suppression effects on expression of inflammatory factors by LY2874455 were in a wide-range manner, as it also showed suppressive effects upon polyI:C treatment (**S2A–S2C Fig**), bacteria treatment (**S2D–S2F Fig**), and $H_2O_2$ treatment (**S2G–S2K Fig**). LY2874455 also inhibited inflammasome-mediated cytokine (IL18) induction (**Fig 2L**), which was in line with previous studies on Inflammasome regulation by autophagy.

## LY2874455 reduced the protein levels of immunoproteasomes in LPS-stimulated macrophages

To determine the mechanism by which LY2874455 suppresses inflammatory responses, mass spectrometry-based proteomics analysis was performed on RAW264.7 macrophages treated with LPS and LY2874455 (S2 Table and **Fig 3A**, upper panel). Clustering of the altered proteins suggested that the immunoproteasome-specific subunits were down-regulated by LY2874455 treatment (arrow pointed out in **Fig 3A**, lower panel). Under inflammatory conditions such as LPS stimulation, the immunoproteasome-specific subunits β1i/LMP2, β2i/MECL-1/LMP10, and β5i/LMP7 replaced the original β1, β2, and β5 subunits of the 20s proteasome, leading to the formation of immunoproteasomes (schemed in **Fig 3B and 3C**). The function of the immunoproteasome in NF-κB pathway was confirmed by the observation that

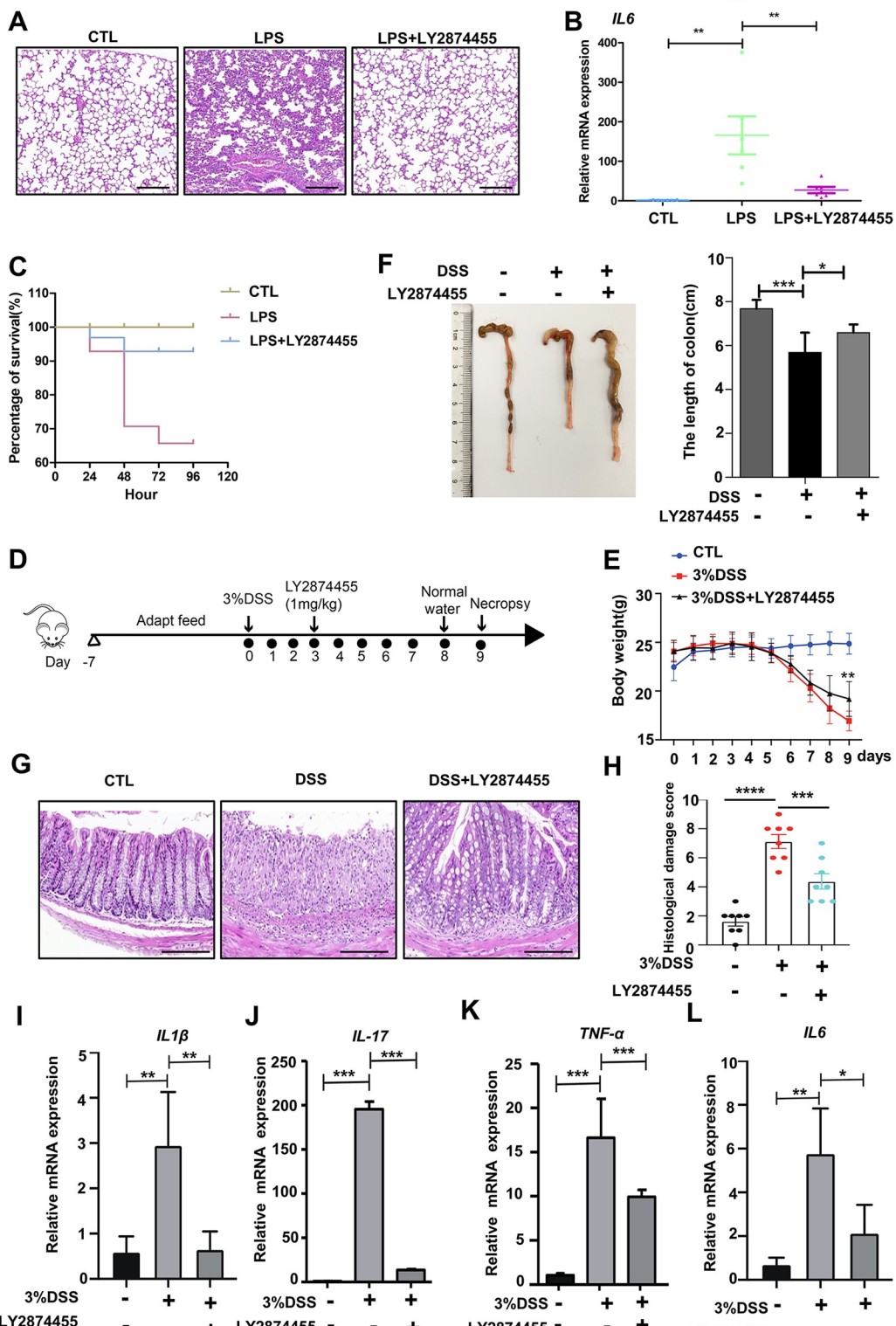

**Fig 2. LY2874455 suppressed lung and intestine inflammation in mouse models. (A)** LY2874455 reduced immune cell infiltration in lung tissues from acute lung injury mouse models induced by LPS. Two-month-old male C57BL/6J mice were subject to intragastric administration with LY2874455 (1 mg/kg) and/or intraperitoneal injection with LPS (10 mg/kg) for 6 h (*n* = 6). Afterwards, upper lobe of right lungs was collected for HE staining and representative images were displayed. Scale bar: 100 μm. **(B)** LY2874455 suppressed the expression of inflammatory cytokine IL-6 in acute lung

injury mouse models induced by LPS. Two-month-old male C57BL/6J mice were treated as in (A), and relative mRNA levels of IL-6 in lung tissues (lower lobe of right lungs) were analyzed by qRT-PCR ($n = 6$). **(C)** Two-month-old male C57BL/6J mice were subject to intragastric administration with LY2874455 (1 mg/kg) and/or intraperitoneal injection with LPS (35 mg/kg) ($n = 8$). The survival test results were presented as Kaplan–Meier survival curves. LY2874455 reduced inflammatory injury in intestine tissues from DSS salt-induced inflammatory bowel mouse models. **(D)** Two-month-old male C57BL/6J mice were subject to intragastric administration with 3% DSS for 7 days and with LY2874455 (1 mg/kg) from the third day ($n = 8$). **(E)** The body weight was recorded and quantified. **(F)** The length of the colons was shown by representative pictures and quantification data. **(G)** LY2874455 reduced intestine injury caused by DSS-induced colitis. Colonic tissues from indicated mice were subject to HE staining. Representative images were shown ($n = 8$). Scale bar: 100 μm. **(H)** The histological score was evaluated. **(I–L)** LY2874455 reduced the levels of inflammatory cytokine genes induced in DSS-induced mouse models. The expression of inflammatory cytokine genes IL-17 and TNF-α was checked by qRT-PCR in colons from indicated mice ($n = 8$). *: $P < 0.05$, **: $P < 0.01$, ***: $P < 0.001$, ****: $P < 0.0001$, NS: no statistical difference. The data underlying the graphs shown in the figure can be found in S1 Data. DSS, dextran sulfate sodium; HE, hematoxylineosin; LPS, lipopolysaccharide; qRT-PCR, quantitative real-time polymerase chain reaction.

immunoproteasome inhibitor ONX-0914 efficiently blocked the nucleus translocation of p65 (**S3A Fig**). The function of immunoproteasome in NF-κB pathway was suggested to be not through degrading p-IκB-α (**S3B Fig**). Instead, immunoproteasome may function in NF-κB pathway through regulating the upstream signaling steps that promotes the phosphorylation of IκB-α (**S3C Fig**). The exact mechanism of immunoproteasome in regulating the NF-κB pathway has been elusive. We analyzed the NF-κB pathway in the presence of LY2874455 or ONX-0914 by analyzing the components function in NF-κB pathway. LY2874455 or ONX-0914 caused the reduction of protein levels of TRAF6 and TAK1 (**S3D and S3E Fig**). These results suggested that immunoproteasome may regulate the TRAF6-TAK1 phase in NF-κB pathway. Further, the same effects of LY2874455 and ONX-0914 suggested that LY2874455 suppressed the inflammation pathway through immunoproteasome.

We confirmed the proteomics results and found that LY2874455 indeed reduced the protein levels of the immunoproteasome subunits LMP2, LMP7, and LMP10 in LPS-stimulated RAW264.7 macrophages (**Fig 3D**). In contrast, the protein levels of constitutive proteasome subunits (PSMB1, PSMB5, PSMB6, and PSMB7) in LPS-stimulated RAW264.7 macrophages were not affected by LY2874455 (**Figs 3E** and **S3G**). Similar results were also obtained in PMs isolated from mice (**Fig 3F and 3G**). Moreover, the catalytic activities of immunoproteasome subunits (LMP2, LMP7, and LMP10) in LPS-stimulated RAW264.7 macrophages were also reduced by LY2874455 (**Figs 3H** and **S3F**). ONX-0914 has been reported as the specific inhibitor of the immunoproteasome [71]. We confirmed that ONX-0914 effectively suppressed the inflammatory responses in LPS-stimulated RAW264.7 macrophages by measuring the expression of the proinflammatory cytokines TNF-α and IL-6 (**Fig 3I and 3J**). More importantly, additional treatment with LY2874455 did not further repress expression of proinflammatory factors in ONX-0914-treated macrophages (**Fig 3I and 3J**). This result indicated that LY2874455 repressed expression of proinflammatory factors, which was speculated to be mediated by lowering the protein levels of immunoproteasomes.

## LY2874455 activated autophagy

We then tried to determine the mechanism by which LY2874455 reduces the protein levels of immunoproteasome subunits (equivalent to immunoproteasomes). The ubiquitin–proteasome and autophagy–lysosome systems have been reported to be the 2 main mechanisms of cytoplasmic degradation [139]. In the 2 intracellular degradation pathways (the proteasome pathway and the autophagy pathway), large substrates are generally targeted for autophagic degradation. We blocked these 2 degradation pathways using proteasome inhibitor MG132 and autophagy inhibitors chloroquine (CQ) and wortmannin (targeting Class III PI3K, VPS34

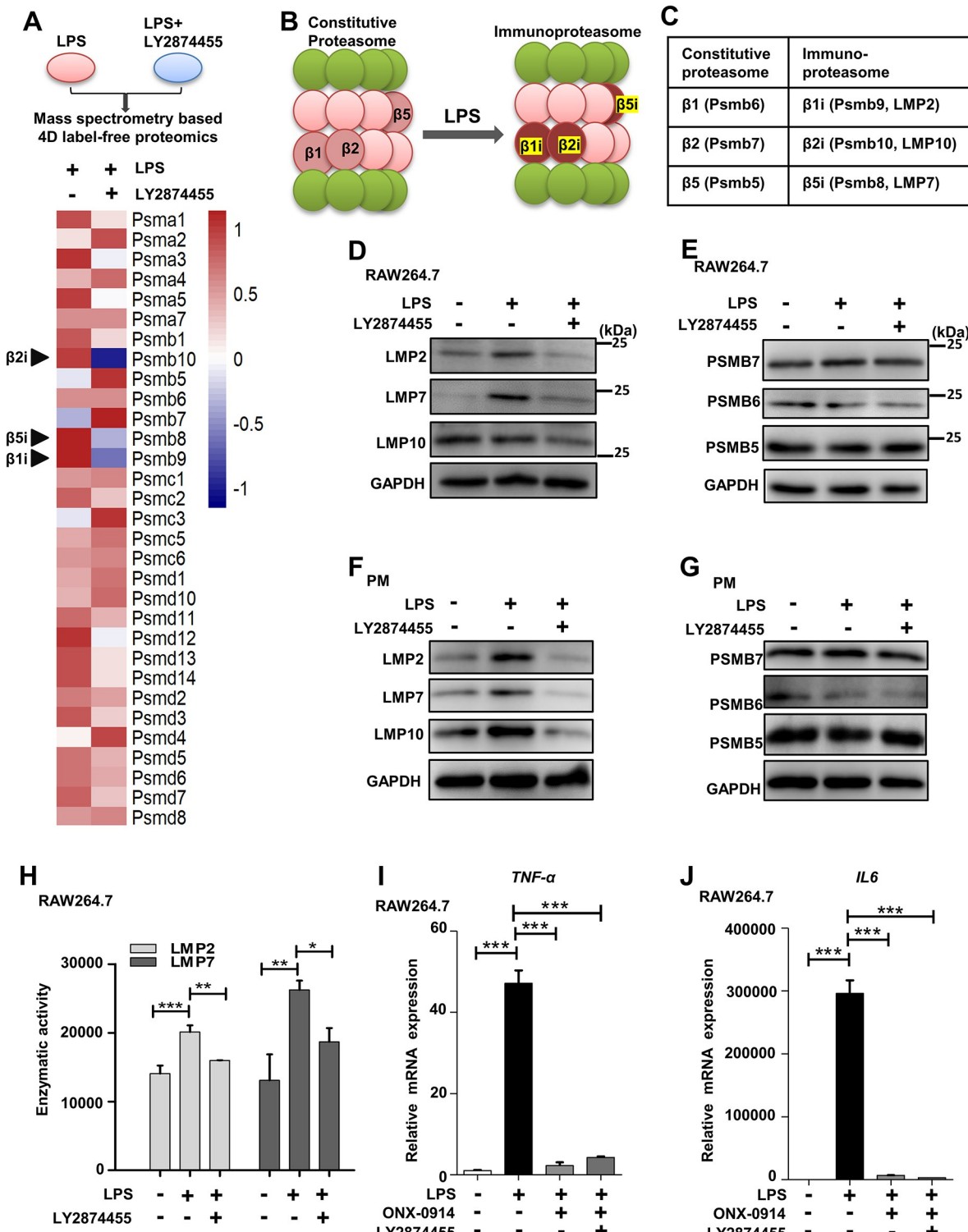

**Fig 3. LY2874455 reduced the protein levels of immunoproteasome-specific subunits in LPS-stimulated macrophages. (A)** Above, method diagram of mass spectrometry-based proteomics analysis of RAW264.7 cells treated with LPS (20 ng/ml) or/and LY2874455. The original mass spectrometry proteomics data have been deposited to the ProteomeXchange Consortium via the PRIDE partner repository with the dataset identifier PXD038747. Bottom, heat map of protein levels of proteasome subunits. The protein levels of immunoproteasome-specific subunits (arrow pointed) were specifically reduced by LY2874455 treatment. **(B, C)** The diagram and names of the specific subunit of 20s constitutive proteasome and immunoproteasome. **(D)** LY2874455 reduced the protein levels of

immunoproteasome specific subunits. Protein levels of immunoproteasome subunits LMP2, LMP7, LMP10, and control GAPDH were analyzed in RAW264.7 cells treated with LY2874455 (2 μm) and LPS (20 ng/ml) for 24 h. The blots were quantified with densitometric values and statistical significance was analyzed in S3 Fig. (E) LY2874455 had no effect on the protein levels of constitutive proteasome-specific subunits. Protein levels of constitutive proteasome subunits PSMB5, PSMB6, PSMB7, and control GAPDH were analyzed. The blots were quantified with densitometric values and statistical significance was analyzed. (F) PMs from C57BL/6J mice were obtained and treated with LPS (20 ng/ml) and/or LY2874455 (2 μm) for 24 h. The total proteins were extracted to detect the protein levels of immunoproteasome subunits LMP2, LMP7, LMP10, and GAPDH. The blots were quantified with densitometric values and statistical significance was analyzed. (G) Protein levels of constitutive proteasome subunits PSMB5, PSMB6, PSMB7, and GAPDH were detected. The blots were quantified with densitometric values and statistical significance was analyzed. (H) The enzymatic activities of immunoproteasome subunits LMP2 and LMP7 were measured in RAW264.7 cells. *: $P < 0.05$, **: $P < 0.01$, ***: $P < 0.001$. (I, J) LY2874455 could not cause further repression on inflammation in macrophages with inhibited immunoproteasome. The specific inhibitor of immunoproteasome, ONX-0914, suppressed expression of inflammatory cytokines in macrophages stimulated with LPS. Additional treatment with LY2874455 could not cause further inflammation-repression. The expression of inflammatory cytokine genes (TNF-α and IL-6) was checked by qRT-PCR in RAW264.7 cells treated with LPS, the immunoproteasome inhibitor ONX-0914(1 μm) and LY2874455 (2 μm) for 24 h. ***: $P < 0.001$. NS: no statistical difference. The data underlying the graphs shown in the figure can be found in S1 Data. LPS, lipopolysaccharide; PM, peritoneal macrophage; qRT-PCR, quantitative real-time polymerase chain reaction.

that uses PtdIns as a substrate to produce PtdIns3P, a lipid material for autophagosome formation [140,141]), and then analyzed the protein levels of immunoproteasome subunits. Proteasome inhibitor MG132 and autophagy inhibitors CQ and wortmannin were checked for their effects on cell viability and the concentrations without affecting cell viability were used (S4A Fig). Western blot analysis showed that MG132 cannot inhibit the LY2874455-induced reductions in LMP2 and LMP7 expression in PM or in RAW264.7 (Figs 4A and S4B), whereas the decrease induced by LY2874455 was markedly reversed by CQ or wortmannin (Figs 4B, 4C and S4C). This result thus suggested that immunoproteasome degradation induced by LY2874455 was more dependent on autophagy than the ubiquitin–proteasome system.

These findings also suggested that LY2874455 can induce autophagy. Autophagic flux assays were conducted by measuring the dot signal of the fusion protein mCherry-EGFP-LC3 [142]. GFP signal is quenched upon autophagosome-lysosome fusion, but the mCherry signal is not affected, therefore the red signal indicates autophagy activation. The high intensity of the mCherry signal and low GFP signal induced by LY2874455 indicated that LY2874455 activated autophagy (Fig 4D). Transmission electron microscopy showed many more autophagosomes in RAW264.7 macrophages treated with LY2874455 than in control cells (Fig 4E). p62/SQSTM1 is an autophagic receptor that recruits substrates (usually ubiquitinated substrates) into autophagosomes, and it is degraded together with the substrates after the autophagosome fuses with the lysosome [143]. The protein levels of autophagy substrate p62 were reduced with the addition of LY2874455, while the levels of the autophagosome marker LC3-II were increased by LY2874455 treatment (Fig 4F). Autophagy inhibitor Bafilomycin A1 reversed the effects of LY2874455 (Fig 4F). These results suggested the activation of autophagy by LY2874455. LPS stimulation caused the accumulation of p62 (Fig 4G), indicating that LPS blocked autophagy, which was consistent with the observation that LPS reduced autophagic flux (Fig 4D, yellow signal in LPS cells). LY2874455 effectively reduced the protein levels of p62 in LPS-stimulated RAW264.7 macrophages (Fig 4G). In PMs and BMDMs isolated from mice, LY2874455 also reduced the protein levels of p62 (Fig 4H and 4I). These results confirmed the role of LY2874455 in activating autophagy.

FGFRs are transmembrane RTKs and activated by FGF to form dimers, leading to the activation of downstream FRS2 complex and subsequent activation of PI3K/AKT/mTOR signaling pathway [144–147]. mTOR targets ULK1 but inhibits the activation of the complex through inhibitory phosphorylation [141,148]. We found that LY2874455 reduced the levels of AKT-mTOR signaling, as shown by the low levels of mTOR and AKT phosphorylation upon LPS-induced activation (Fig 4J). This LY2874455-mediated inhibition of the AKT-mTOR signaling pathway further indicated its ability to support activation through inhibition of AKT-

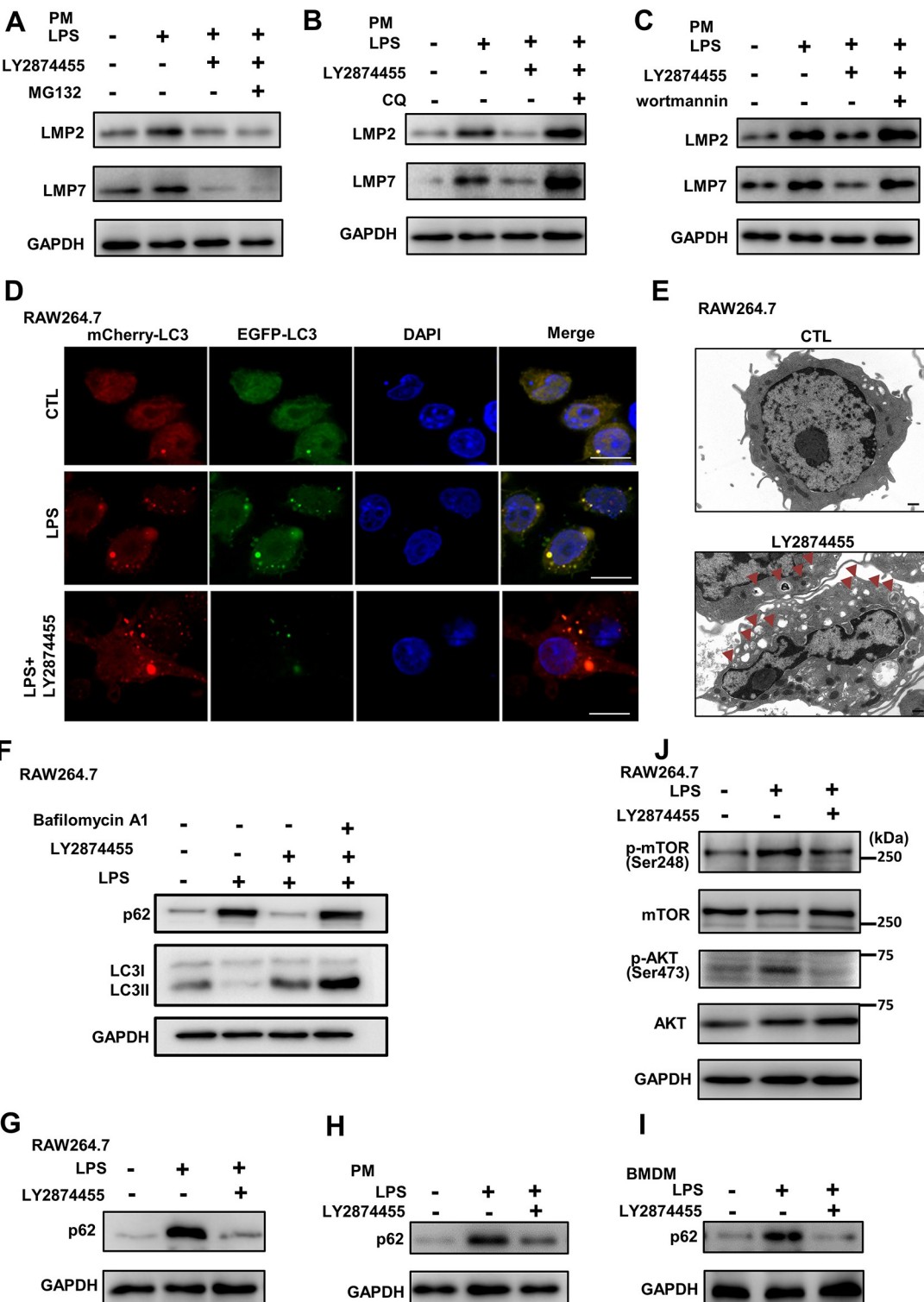

**Fig 4. LY2874455 activated the autophagy process.** (**A**) Proteasome inhibitor MG132 cannot block the reduction of immunoproteasome subunits induced by LY2874455. Protein levels of LMP2, LMP7, and GAPDH were analyzed in RAW264.7 cells treated with LPS (20 ng/ml), LY2874455 (2 µm), and MG132 (5 µm). The blots were quantified with densitometric values and statistical significance was analyzed. (**B, C**) Autophagy inhibitors blocked the reduction of immunoproteasome subunits induced by LY2874455. Protein levels of LMP2, LMP7, and GAPDH were analyzed in RAW264.7 cells treated with LPS (20 ng/ml), LY2874455 (2 µm), and autophagy inhibitors chloroquine (CQ, 20 µm) or wortmannin (100 nM). The blots were

quantified with densitometric values and statistical significance was analyzed. (D) LY2874455 enhanced autophagy flux. RAW264.7 cells transfected with mCherry-EGFP-LC3 were stimulated with LPS (20 ng/ml) or/and LY2874455 for another 24 h, and the puncta of LC3 in cells was monitored after fixation under a confocal fluorescent microscope. Higher level of mCherry-LC3 signal and lower level of EGFP-LC3 signal (showing red in merge) indicated the stimulated autophagy flux, while the similar signal levels of mCherry-LC3 and GFP-L3 (showing yellow in merge) indicated the blocked autophagy flux. Representative images from 3 replicated experiments were shown. Scale bars: 10 μm. (E) LY2874455 promoted the formation of autophagosomes. RAW264.7 cells treated with control or LY2874455 (2 μm) for 24 h were analyzed by transmission electron microscopy. Autophagosomes were pointed by arrows and representative images from 3 replicated experiments were shown. Scale bars: 1 μm. (F) LY2874455 promoted the autophagic degradation of substrate receptor p62. RAW264.7 cells were treated with LPS (20 ng/ml) and LY2874455 (0.5, 2, 4 μm) with or without Bafilomycin A1 (20 nM) for 24 h and protein levels of p62, LC3, and GAPDH were analyzed by western blot. The blots were quantified with densitometric values and statistical significance was analyzed. (G) Protein levels of p62 and GAPDH were detected by western blot in RAW264.7 cells treated with LY2874455 (2 μm) and LPS (20 ng/ml) and for 24 h. The blot was quantified with densitometric values and statistical significance was analyzed. (H) Protein levels of p62 and GAPDH were detected by western blot in PMs from C57BL/6J mice treated as in (G). The blot was quantified with densitometric values and statistical significance was analyzed. (I) Protein levels of p62 and GAPDH were detected by western blot in BMDMs from C57BL/6J mice treated as in (G). The blot was quantified with densitometric values and statistical significance was analyzed. (J) LY2874455 inhibited the AKT-mTOR signaling. Protein levels of p-mTOR (Ser248), mTOR, p-AKT (Ser473), AKT, and GAPDH were detected by western blot in RAW264.7 cells treated as in (G). The blots were quantified with densitometric values and statistical significance was analyzed. *: $P < 0.05$, **: $P < 0.01$, ***: $P < 0.001$, ****: $P < 0.0001$, NS: no statistical difference. BMDM, bone marrow-derived macrophage; LPS, lipopolysaccharide; PM, peritoneal macrophage.

mTOR signaling. LY2874455 suppresses the FGFR activities and the downstream PI3K/AKT/mTOR signaling pathway, leading to the release and activation of autophagy (schemed in **S4D Fig**). In fibroblast cells, LY2874455 also reduced the protein levels of the autophagic substrate p62 while increased the protein levels of the autophagosome marker LC3-II (**S4E Fig**). Another FGFR inhibitor ADZ4547 also reduced the protein levels of p62, LMP2, and LMP7 and inhibited the expression of inflammatory factors (**S4F–S4I Fig**). Other chemicals in our screen that inhibited the expression of proinflammatory factors induced by LPS (**S1B Fig**) showed either promotion or inhibition on the autophagic degradation of p62 (**S4J Fig**), suggesting the effects of LY2874455 on suppression of expression of proinflammatory factors and autophagy induction were specific.

## LY2874455 promoted the degradation of immunoproteasomes through autophagy and the selective receptor p62

For further validation of the effect of LY2874455 on promoting the degradation of immunoproteasomes through autophagy, ATG7, an essential gene in autophagy, was knocked out in RAW264.7 cells, which resulted in blocked autophagy, an increase in p62 and deficiency in the activated form of LC3 (LC3-II) (**S5A Fig**). Then, the protein levels of the immunoproteasome subunits LMP2 and LMP7 were analyzed. In ATG7-knockout (ATG7-KO) RAW264.7 cells, the protein levels of LMP2 and LMP7 were increased compared to those in WT cells (**Fig 5A**). While the protein levels of LMP2 and LMP7 were reduced by LY2874455 in WT cells, this effect was abolished in ATG7-KO cells (**Fig 5B and 5C**). We examined the localization of autophagosomes and the immunoproteasome. The results showed that an autophagosome marker (LC3 dots) colocalized with LMP2 and LMP7 (**Fig 5D**). Collectively, these results demonstrated that LY2874455 induced the selective capture and degradation of immunoproteasomes via autophagy.

Generally, selective targeting of substrates into autophagosomes is mediated by receptors that link cargos to autophagosomes and are degraded together with the substrates after autophagic transfer into lysosomes. We then analyzed 3 previously reported receptors and found that p62 behaved similarly to LMP2 and LMP7 in RAW264.7 cells, was increased by LPS treatment and decreased by further LY2874455 treatment (**Fig 5E**). Knockdown of p62 in RAW264.7 cells led to increased protein levels of LMP2 and LMP7 (**Figs 5F** and **S5B Fig**).

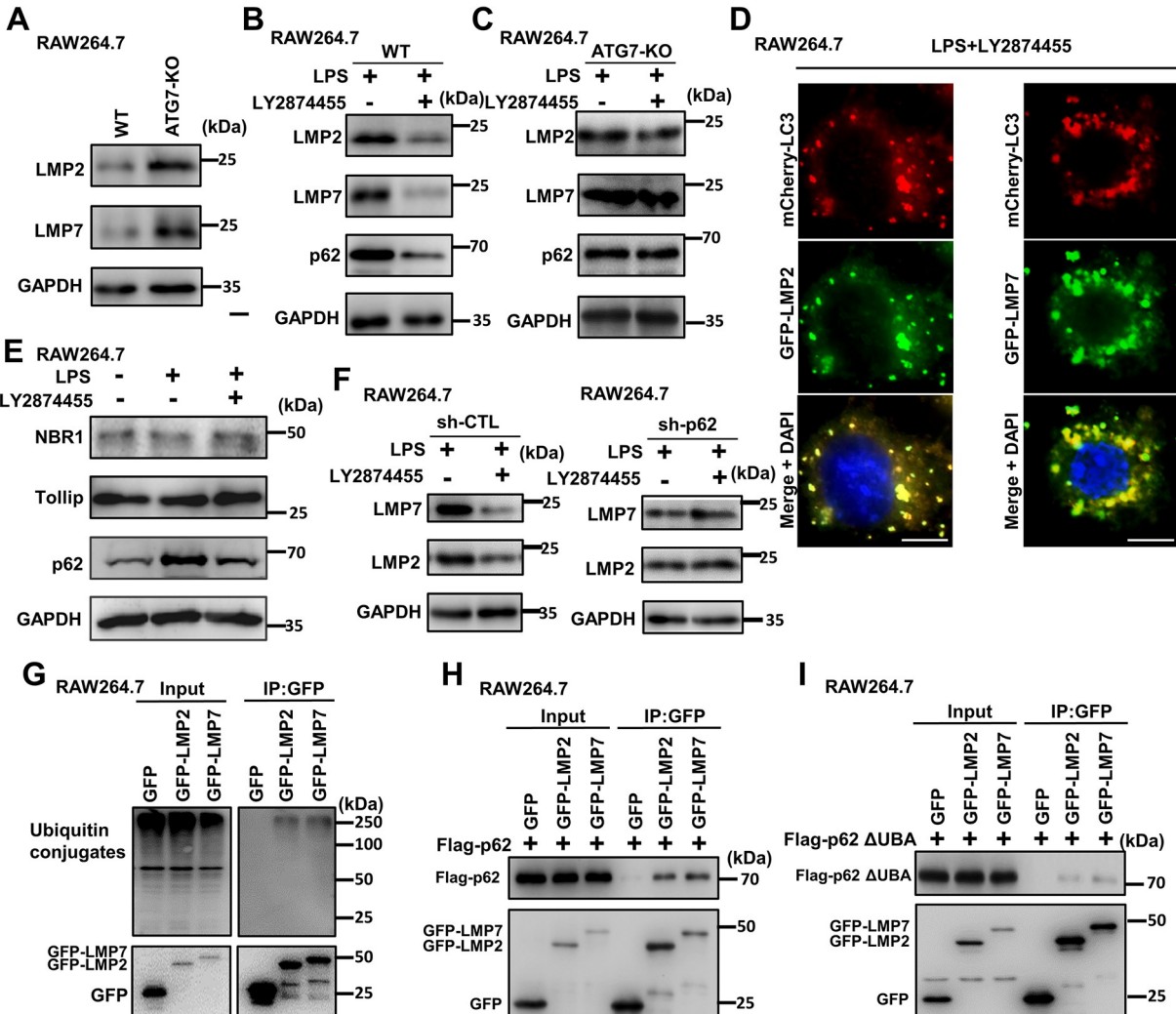

**Fig 5. LY2874455 promoted the degradation of immunoproteasomes through autophagy and selective receptor p62. (A)** Autophagy deficiency caused increased levels of immunoproteasome subunits. Protein levels of LMP2, LMP7, and GAPDH were detected by western blot in WT or ATG7 knock out (ATG7-KO, through CRISPR-CAS9 method) RAW264.7 cells. The blots were quantified with densitometric values and statistical significance was analyzed. **(B, C)** LY2874455 could not reduce the protein levels of immunoproteasome subunits in autophagy-deficient macrophages. WT or ATG7-KO RAW264.7 cells were treated with LPS (20 ng/ml) alone or together with LY2874455 (2 μm) for 24 h. Protein levels of LMP2, LMP7, p62, and GAPDH were analyzed by western blot. The blots were quantified with densitometric values and statistical significance was analyzed. **(D)** Immunoproteasome subunits colocalized with autophagosome marker LC3. RAW264.7 cells expressing mCherry-tagged LC3 and GFP-tagged LMP2 or LMP7 were stimulated with LPS (20 ng/ml) and LY2874455 (2 μm) for 24 h. The colocalization of GFP-LMP2/GFP-LMP7 with mCherry-LC3 was observed by confocal fluorescent microscope. Scale bars: 10 μm. **(E)** LY2874455 reduced the protein levels of selective autophagy receptor p62. RAW264.7 cells were treated with LPS (20 ng/ml) and LY2874455 (2 μm) for 24 h, and then the protein levels of NBR1, Tollip, p62, and GAPDH were detected by western blot. The blots were quantified with densitometric values and statistical significance was analyzed. **(F)** Knockdown of p62 caused increase of protein levels of immunoproteasome subunits. The protein levels of LMP2, LMP7, and GAPDH were detected by western blot in control shRNA or p62-targeting shRNA transfected RAW264.7 cells treated with LPS (20 ng/ml) and LY2874455 (2 μm) for 24 h. The blots were quantified with densitometric values and statistical significance was analyzed. **(G)** Immunoproteasome subunits were ubiquitinated. RAW264.7 cells expressing GFP-LMP2 or GFP-LMP7 were treated with LY2874455 (2 μm) for 24 h. Then, equal amounts of protein lysates were subject to immunoprecipitation with anti-GFP affinity beads and analyzed by western blot with anti-GFP and anti-ubiquitin antibodies. **(H, I)** p62 interacted with immunoproteasome subunits through its ubiquitin-binding UBA domain. RAW264.7 cells expressing GFP-LMP2, GFP-LMP7, and Flag-tagged full-length p62 or truncated p62 (ubiquitin-binding domain UBA deleted) were subject to immunoprecipitation with anti-GFP affinity beads and then analyzed by western blot with anti-GFP and anti-Flag antibodies. *: $P < 0.05$, **: $P < 0.01$, ***: $P < 0.001$, ****: $P < 0.0001$, NS: no statistical difference. LPS, lipopolysaccharide; WT, wild type.

These results indicated p62 was the receptor protein for the autophagic degradation of immunoproteasomes. Substrate ubiquitination is necessary for selective targeting by the autophagy receptor p62. Indeed, LMP2 and LMP7 were ubiquitinated (**Fig 5G**). Furthermore, LMP2 and LMP7 could interact with p62, and this interaction was shown to be dependent on the UBA domain (ubiquitin-associated domain for ubiquitin binding) of p62 (**Fig 5H and 5I**). Thus, these results indicated p62 was the receptor that mediated the autophagic degradation of immunoproteasomes.

### LY2874455 suppressed expression of proinflammatory factors though the autophagy pathway

Now that we have shown that LY2874455 induced the autophagic degradation of immunoproteasomes, we then examined whether the suppressive effect of LY2874455 on expression of proinflammatory factors occurred through the autophagy pathway. The LPS-induced increase in ROS levels (shown by the DCFH-DA probe) in RAW264.7 cells was abolished by LY2874455; however, this abolishment was blocked by the autophagy inhibitor wortmannin (**Fig 6A**). Moreover, increased mRNA levels of iNOS and IL-6 were observed in wortmannin-treated RAW264.7 cells compared to LY2874455-treated RAW264.7 cells (**Fig 6B and 6C**), mouse-isolated primary macrophages (**Fig 6D and 6E**), and BMDMs (**S6A and S6B Fig**). These results suggested that LY2874455 suppressed expression of proinflammatory factors through induction of autophagy. This finding was further confirmed by the observation that ATG7 knockout or p62 knockdown led to increased levels of iNOS and IL-6 in the presence of LPS and LY2874455 (**Fig 6F and 6G**). The effects of LY2874455 suppression on NF-κB pathway was also shown to be dependent on autophagy and receptor p62, as in autophagy- (**S6C–S6F Fig**) or p62-deficient (**S6G and S6H Fig**) macrophages, LY2874455 cannot suppress NF-κB pathway. Autophagy inhibitors reversed the effects of LY2874455 (**S6J and S6K Fig**), further indicating that LY2874455 suppressed expression of proinflammatory factors through autophagy.

Together with the data showing that LY2874455 induced the autophagic degradation of immunoproteasomes, these results indicated the possibility that the increased expression of proinflammatory factors in autophagy-deficient cells may be caused by the accumulation of autophagic substrate immunoproteasomes. The immunoproteasome-specific inhibitor ONX-0914 was then used to treat ATG7-KO RAW264.7 cells, and the expression levels of iNOS and the proinflammatory cytokines IL-6 and TNF-α were analyzed. The results showed that the immunoproteasome inhibitor ONX-0914 dramatically reduced expression of proinflammatory factors in autophagy-deficient cells (**Figs 6H–6J** and **S6I Fig**), indicating that the activated immunoproteasome in autophagy-deficient cells induced expression of proinflammatory factors. In LMP7 knockdown cells, ONX-0914 did not inhibit the expression levels of inflammatory factor (**S6L and S6M Fig**), indicating the specific effects of ONX-0914 on immunoproteasome.

Hence, these results indicated that the suppressive effect of LY2874455 on expression of proinflammatory factors was dependent on the induction of the autophagic degradation of immunoproteasomes.

## Discussion

It has been found that disruption of autophagy leads to diseases with inflammatory components, including infections, autoimmunity and metabolic disorders [149–153]. Autophagy restricts inflammation through various mechanisms by targeting inflammatory mediators including inflammasomes, STING and MAVS for cytosolic pathogen-associated molecular

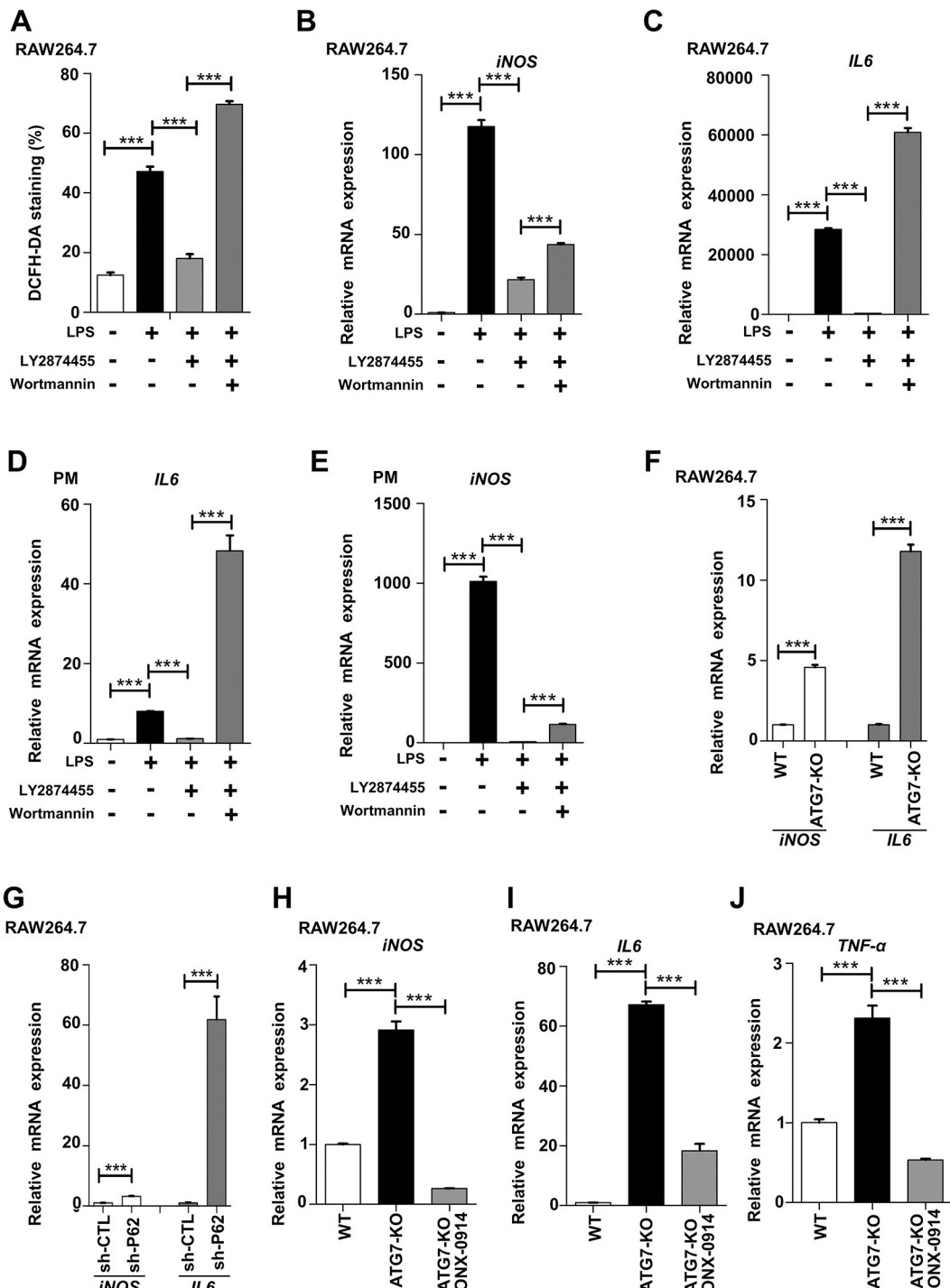

**Fig 6. LY2874455 suppressed inflammation though autophagy pathway. (A)** Autophagy inhibitor reversed the effect of LY2874455 on ROS-reduction in LPS-stimulated macrophages. RAW264.7 cells were stained with ROS-probe DCFH-DA after treatment with LPS (20 ng/ml), LY2874455 (2 μm), and wortmannin (100 nM) as indicated for 24 h. Quantitative measurement of cellular ROS was performed by flow cytometry. **(B, C)** Autophagy inhibitor reversed the effect of LY2874455 in LPS-stimulated macrophages. The expression of inflammatory cytokine genes (iNOS and IL-6) was checked by qRT-PCR in RAW264.7 cells treated as (A). **(D, E)** The expression of inflammatory cytokine IL-6 and iNOS was checked by qRT-PCR in PMs as treated as in (A). **(F)** Autophagy deficiency reversed the effect of LY2874455 in inflammatory macrophages. WT or ATG7 knock out (ATG7-KO) RAW264.7 cells were treated with LPS (20 ng/ml) and LY2874455 (2 μm) for 24 h. The expression of iNOS and IL-6 was checked by qRT-PCR and expressed as fold changes compared to values in WT cells. **(G)** p62 knock down reversed the effect of LY2874455 in inflammatory macrophages. Control (sh-CTL,

control shRNA) or p62 knockdown (sh-p62, p62-targeting shRNA) RAW264.7 cells were treated with LPS (20 ng/ml) and LY2874455 (2 μm) for 24 h. The expression of iNOS and IL-6 was checked by qRT-PCR and expressed as fold changes compared to values in control cells. **(H–J)** The enhanced production of proinflammatory factors in autophagy-deficient macrophages was suppressed by immunoproteasome inhibitor ONX-0914. WT or ATG7 knockout (ATG7-KO) RAW264.7 cells treated with LPS (20 ng/ml) and ONX-0914 (1 μm). The expression of iNOS and inflammatory cytokine genes (IL-6 and TNF-α) was checked by qRT-PCR. *: $P < 0.05$, **: $P < 0.01$, ***: $P < 0.001$, ****: $P < 0.0001$, NS: no statistical difference. The data underlying the graphs shown in the figure can be found in S1 Data. iNOS, inducible nitric oxide synthase; LPS, lipopolysaccharide; PM, peritoneal macrophage; qRT-PCR, quantitative real-time polymerase chain reaction; ROS, reactive oxygen species; WT, wild type.

pattern (PAMP) recognition [154–162]. In the present study, we uncovered the role and mechanism of the pan-FGFR inhibitor LY2874455 in suppression of production of proinflammatory factors. In macrophages stimulated with LPS, immunoproteasomes were induced and then triggered NF-κB-mediated inflammatory responses, leading to robust expression and generation of proinflammatory cytokines, ROS, and NO. In the presence of LY2874455, the autophagy pathway was activated and captured immunoproteasomes, and this effect was mediated by the selective receptor p62 recognizing ubiquitin signals on immunoproteasomes. This autophagic degradation reduced the abundance of immunoproteasomes and thus blocked the activity of NF-κB and eventually suppressed inflammation. Our work here reveals a novel mechanism by which autophagy suppresses inflammation, which may substantially contribute to the development of new therapeutic approaches for inflammatory diseases.

Currently, it is unclear how the induction and autophagic degradation of immunoproteasomes are coordinated. We hypothesize that inflammatory stimuli such as LPS may induce the expression, assembly, and increased abundance of immunoproteasomes, while LPS inhibits the autophagy pathway, as shown by the increased protein levels of the receptor p62 (**Fig 4F–4I**). This result suggests that LPS not only induces the production of the immunoproteasome but also prohibits its autophagic degradation, which consequently causes the accumulation of the immunoproteasome and eventual rampant inflammation. The high level of immunoproteasome-induced inflammation provides efficacious signals to remind the host of toxic stimuli and thus activate host responses [57]. However, the host benefits from intense inflammation-induced signals and simultaneously suffers from inflammation-induced damage, especially under severe/acute or chronic inflammatory conditions. Thus, the balance between moderate and controllable inflammation is essential in maintaining host health. Here, we propose a mechanism to explain how autophagy functions to break rampant inflammation. The selective autophagic degradation of the immunoproteasome in macrophages suppressed the production of proinflammatory factors (**S6R Fig**).

LY2874455, a novel pan-FGFR inhibitor, was shown to have good tolerability, absorbance, and activity in solid organ cancer patients in a Phase I clinical study [130]. Thus, LY2874455 could be a good therapeutic candidate for inflammation-related diseases such as sepsis and Crohn's disease.

In addition to the regulation of acute inflammation, autophagy has been shown to positively alleviate chronic inflammation-related neurodegenerative disorders such as Alzheimer's disease, Parkinson's disease, and depression [163,164]. An overwhelming number of studies have connected autophagy in neurons to degenerative diseases; however, there are only sporadic hints of inflammatory connections via autophagy. Recently, it was shown that LPS could impair autophagy and relieve autophagic inhibition of the inflammatory machinery in microglia, a specific type of macrophage in the central nervous system [165–167]. Autophagy-deficient microglia show impaired phagocytic capacities and the overproduction of proinflammatory cytokines, which can lead to synaptic dysfunction, neuronal death, and the inhibition of neurogenesis [168]. It will be interesting to examine whether autophagic

degradation of the immunoproteasome shows anti-inflammatory effect in microglia and whether and how much this dysregulation contributes to neurodegenerative diseases.

Cautiously, while in some cases defective autophagy causes excessive inflammation that leads to disease pathology, there are certain situations in which elevated inflammation promotes survival of model animals. For example, myeloid-specific knockout of autophagy genes (RB1CC1, ATG5, ATG7, or ATG14) caused increased inflammation in the lungs and facilitated resistance to influenza virus infection in animals [169]. Additionally, myeloid-specific knockout of several autophagy genes (RB1CC1, Beclin1, ATG3, ATG5, ATG7, ATG14, or ATG16L) in animals subjected to herpesvirus infection results in elevated inflammation and prevented viral reactivation from latency [170]. We hypothesize that for pathogens that induce normal (slight increase above baseline) inflammation, autophagy deficiency results in elevated inflammation that is beneficial for host recognition and reaction to invaded pathogens, while for pathogens that induce high levels of inflammation, autophagy deficiency further increases inflammation, and this inflammation leads to damage to the host. That is, although we showed that the autophagy-mediated reduction in the immunoproteasome is beneficial for animals with LPS-induced lung damage and DSS-induced inflammatory bowel damage, it is possible that in certain situations, autophagic degradation of the immunoproteasome may weaken proper host responses to pathogens. Therefore, whether and how autophagic degradation of the immunoproteasome functions in inflammation-related diseases such as aging, cancer, cardiovascular disease, neurodegeneration, and infections needs to be specifically examined.

The functions of selective autophagy in various conditions occur through receptors that specifically recognize certain types of substrates and concomitantly interact with lipidated LC3/Atg8 residing in the autophagosome membrane [171,172]. The degradation of substrates mostly indicates specific roles of autophagy [173]. The nature of autophagic substrates varies according to cell type and stimulating signals [110,174]. In our case, selective targeting of the immunoproteasome by autophagy explains the suppressive effect of autophagy on inflammation in macrophages. This conclusion was strengthened by the observation that autophagy blockade induced the accumulation of immunoproteasomes accompanied by intense inflammatory reactions that could be completely reversed by the immunoproteasome inhibitor (**Fig 6H–6J**). This finding suggests that, at least in the contexts examined in this work, autophagic degradation of immunoproteasomes is mainly responsible for the role of autophagy in inflammation repression.

The receptor mediating autophagic degradation of the immunoproteasome is p62 but not NBR1 or Tollip, as shown in this study. One interesting question regarding receptor function in autophagy is why so many receptors exist. For example, p62, NBR1, Tollip, and OPTN function in aggrephagy (autophagy of protein aggregates); BNIP3L, FUNDC1, BNIP3, AMBRA1, BCL2LI3, FKBP8, CHDH, DISC1, PHB2, and cardiolipin function in mitophagy (autophagy of mitochondria); FAM134B, Sec62, RTN3, CCPG1 ATL3, and TEX264 function in ERphagy (autophagy of ER); and p62, NBR1, and PEX3 function in pexophagy (autophagy of peroxisomes) [110,143,174,175]. The redundancy of receptors may reflect the specific and meticulous regulation and function of autophagy in multicellular organisms. In other words, certain receptors functions in specifically defined types of cells. The clarification of where (cell types), when (activation or inhibition), and how (substrate targeting) autophagy functions may improve our knowledge of the intrinsic nature of autophagy as a dynamic and multifunctional process. The observation that immunoproteasome subunits are polyubiquitinated and their binding with p62 relies on the ubiquitin-binding domain of 62 indicates ubiquitin chains as the signal for immunoproteasome targeting by autophagosomes. Identifying the ubiquitin ligase(s) responsible for ubiquitination of the immunoproteasome may help to determine how ubiquitin ligase(s) directly or indirectly facilitate the specific recognition of the

immunoproteasome by p62. Interestingly, p62 and other receptor-mediated autophagy has been found to be responsible for autophagic degradation of constitute proteasome [176–178].

Autophagy not only functions as a degradation pathway, but also functions to facilitate extracellular secretion [179]. In the study on chondrocytes [180], the authors found that autophagy is induced in growth-plate chondrocytes and regulates the secretion of type II collagen (Col2). We speculate that in chondrocytes at growth conditions, autophagy is activated and necessary for cell secretion of certain factor.

Hence, autophagy functions differently in different types of cells. It is interesting that FGF signaling (FGF18 and FGFR4) activates autophagy in chondrocytes while in our case FGF signaling inhibition (FGFR inhibitor LY2874455 treatment) activates autophagy. It is possible that the main functional downstream kinases of FGF signaling may be different in different types of cells, for example, JNK in chondrocytes while AKT and mTOR in macrophages (**Figs 4J** and **S4D**).

In conclusion, our findings described the selective autophagic degradation of the immunoproteasome in the context of aberrant inflammation in macrophages and supported a novel mechanism of the regulation of inflammation by selective autophagy (**S6R Fig**). This study explored the crosstalk between selective autophagy and inflammation, which should be meaningful in establishing effective therapies for inflammatory diseases.

## Materials and methods

### Antibodies and reagents

The following antibodies were purchased from Abcam: anti-SQSTM-1/p62 (ab109012), anti-LMP2 (ab3328), anti-LMP7 (ab3329), anti-Ubiquitin (ab7780), anti-NBR1 (ab55474), anti-ATG7 (ab133528), anti-Mtor (ab87540), anti-LMP10 (ab183506), anti-PSMB5 (ab90867), anti-GAPDH (60004-1-Ig), anti-p65 (10745-1-AP). Anti-PSMB1 (11749-1-AP) and anti-PSMB2 (15154-1-AP) were purchased from Proteintech. Those antibodies anti-LC3 (2775S), anti-phospho-AktSer473 (4060S), anti-phospho-mTORSer2448 (2971S), anti-p70S6K (9202S), anti-phospho-p70S6KThr421/Ser424 (9204S), anti-Akt (4685S), and anti-TNF-α (11948S) were from Cell Signaling Technology. The phospho-NF-kB p65 (Ser536) (bs-0982R) antibody and anti-IL6 (bs-0782R) was purchased from Bioss. Anti-phospho-IκB-α (Ser32) (AP0707), anti-TRIF (A1155), anti-MYD88 (A0980), anti-TRAF6 (A16991), anti-TAK1 (A19077), anti-TAB2 (A9867) antibodies were from ABclonal Technology. Anti-IL1β (ab216995) and anti-IL18 (ab207323) antibodies were from Abcam. The secondary antibodies HRP-conjugated anti-rabbit (D-110058) and HRP-conjugated anti-mouse (D-110087) were from BBI Life Sciences. Fluor-conjugated (488 and 594) secondary antibodies were purchased from Invitrogen. LY2874455, Domiphen (bromide), Cetylpyridinium (chloride monohydrate), Fenticonazole (Nitrate), AZD4547, and ULK1-IN-2 were from MCE and dissolved in phosphate-buffered saline (PBS). LPS (*E. coli*, serotype 055:B5) was purchased from Sigma. The $H_2$-2′,7′-dichloro-dihydrofluorescein diacetate (DCFH-DA) (S0033S) and NO detection Kit(S0021S) were purchased from Beyotime. Protease inhibitor cocktail was purchased from Sigma (Roche 11697498001). Polyinosinic-polycytidylic acid (Poly(I:C)) (HY-107202) was purchased from MCE.

### Macrophage culture and isolation

RAW264.7 macrophages were cultured in RPMI-1640 (Sigma) with 10% fetal bovine serum (Gibco) and maintained in humidified 5% $CO_2$ atmosphere at 37°C. PM and BMDMs were isolated from C57BL/6J mice, and 6 to 8 weeks aged mice were injected with 3% thioglycolate prior to the sacrifice of the mice and the isolation of BMDM. Cells separated from ascites were

seeded in cell plates with RPMI-1640 medium and were allowed to adhere for 4 to 6 h. Adherent cells were considered as PM. Cells from bone marrow were seeded in cell plates incubated with RPMI1640 complete medium with M-CSF (10 ng/ml, Sigma) and were allowed to adhere for 7 days. Adherent cells were considered as BMDM.

## Plasmid construction and DNA transfection

The target gene p62, LMP2, LMP7 were cloned into the pEGFP-C1 vector, respectively. The target gene p62 and p62△UBA were cloned into the pFlag-CMV2 vector, respectively. The target gene LC3 was cloned into the pLVX-mCherry-C1 vector. The sh-RNA targeting p62 sequence (CCGGGTCTCTACAGATGCCAGAATCCTCGAGGATTCTGGCATCTGTAGAGACTT TTTG) was cloned into the PLKO.1 vector. The guide RNA (gRNA) sequence (TCTGCCC ACCCGCTTGACGT) of the target gene ATG7 was cloned into the PX330 vector. Upstream (800 bp) and downstream (800 bp) sequences from the knockout site of the target gene and selection marker (puro) were cloned into the pUC19 vector. Endoxin-free plasmids were transfected into cells using jetprime reagent according to the protocol recommended by the manufacturer (Polyplus).

## ROS detection

For intracellular ROS measurement, the cells treated with LY2874455 and/or LPS for 24 h were washed with PBS 3 times and incubated with 10 μm H2-DCFH-DA for 30 min at 37˚C in the dark. Then, cells were washed 3 times and collected by scraping. Collected cells were resuspended in PBS and their fluorescence intensity corresponding to intracellular DCFH-DA levels was immediately read on a Becton Dickinson FACS. DCFH-DA signals were also visualized using fluorescence microscope (Zeiss Apotome) and quantified with Image J.

## Nitric oxide (NO) concentration measurements

The level of NO derivative nitrite in culture medium was determined by Griess reaction as per the kit's manual (Beyotime). After the reaction, the signal in the supernatant is measured at 540 nm with a multifunctional microplate reader (Biotek).

## Quantitative real-time polymerase chain reaction analysis (qRT-PCR)

Total RNA was extracted from cells lines, primary cells, or tissues by using Fast Pure cell/Tissue Total RNA Isolation Kit (Nanjing Vazyme Biotek). cDNAs were reverse transcribed from 1 μg of RNA using the Superscript II Reverse Transcriptase kit (Nanjing Vazyme). The production was used as templates in quantitative real-time polymerase chain reactions (PCRs) by using SYBR Green reagent (Nanjing Vazyme) according to the protocol recommended by the manufacturer. The sequences of primers used for PCR amplification are shown in S1 Table. GAPDH served as internal normalization control.

## Proteasome activity profiling

Subunit-selective fluorogenic peptide substrates were used to measure the catalytic activities of individual catalytic subunits by monitoring the rate of substrate hydrolysis over time. Briefly, protein lysates were prepared using passive lysis buffer (25 mM Tris (pH 7.5), 100 mM Nacl, 5 mM ATP, 0.2% NP40, 20% Glycerol) and diluted in 20S proteasome assay buffer (20 mM Tris-HCl, 0.5 mM EDTA, 0.035% SDS (pH 8.0)). Enzyme reactions were initiated by the addition of proteasome substrates. Substrates and concentrations were used as following: Ac-ANW-AMC (β5i, 100 μm), Ac-PAL-AMC (β1i activity, 100 μm), and Ac-Arg-Leu-Arg-AMC

(β2i activity, 100 μm). The mixture was incubated at 37˚C for 2 h. Fluorescence signals were measured using a multifunctional microplate reader (Biotek) at the excitation and emission wavelengths of 345 and 445 nm, respectively.

## Immunoblotting

Cells or tissues were lysed with RIPA buffer with protease inhibitors (Roche 11697498001). The protein was quantified by BCA Protein Assay Kit (Sigma) according to the protocol recommended by the manufacturer. After quantified, protein precipitates were subjected to SDS-PAGE. Protein was transferred to PVDF membranes and blocked in 5% non-fat milk in Tris-HCl-buffered saline (TBS) with 0.1% Tween-20 (TBST) for 1 h at room temperature, after which primary antibody (S1 Table) was incubated overnight at 4˚C. Then, the membranes were incubated with the secondary antibody (S1 Table) at room temperature for 1 h. Signals were acquired with a Chemidoc MP Imaging System (Bio-Rad).

## Immunoprecipitation

Cells were lysed in a buffer containing 0.25% TX-100, 1% NP-40, 50 mM Tris-HCl (pH 8), 150 mM NaCl, and 1 mM EDTA. Lysates were adjusted for concentration at 1 to 2 mg/ml and volume and incubated with GFP-Beads (Shenzhen Sports Life Technology Company) at 4˚C overnight. Then, the samples were centrifuged at 2,000 rpm at 4˚C for 3 min. The pellet was washed 5 times with wash buffer A (50 mM tris-HCl (pH 7.5), 100 mM sodium chloride, and 2 mM EDTA). Bound proteins were collected in RIPA buffer (50 mM Tris (pH 7.4), 150 mM NaCl, 0.1% SDS, 1% TritonX-100, and 0.5% sodium deoxycholate), and the precipitated proteins were subjected to SDS-PAGE.

## Immunofluorescence

Cells grown on cover slips were fixed with 4% paraformaldehyde in PBS for 10 min followed by permeabilization with 0.1% TritonX100 for 10 min and incubation with 3% BSA for 1 h at room temperature. Cells were then incubated with primary antibody followed by a PBS wash and incubated with secondary antibody (donkey anti-mouse and anti-rabbit Alexa Fluor 488 or 594). DAPI was used to stain the DNA. Images were captured using the Zeiss LSM 880.

## Mass spectrometry-based proteomics analysis

The original mass spectrometry proteomics data have been deposited to the ProteomeXchange Consortium via the PRIDE partner repository with the dataset identifier PXD038747 and are publicly available as of the date of publication.

Protein extraction. Sample was sonicated 3 times on ice using a high-intensity ultrasonic processor (Scientz) in lysis buffer (8 M urea, 1% Protease Inhibitor Cocktail). The remaining debris was removed by centrifugation at 12,000 g at 4˚C for 10 min. Finally, the supernatant was collected and the protein concentration was determined with BCA kit according to the manufacturer's instructions.

Trypsin digestion. For digestion, the protein solution was reduced with 5 mM dithiothreitol for 30 min at 56˚C and alkylated with 11 mM iodoacetamide for 15 min at room temperature in darkness. The protein sample was then diluted by adding 100 mM TEAB to urea concentration less than 2 M. Finally, trypsin was added at 1:50 trypsin-to-protein mass ratio for the first digestion overnight and 1:100 trypsin-to-protein mass ratio for a second 4-h digestion.

LC-MS/MS analysis. The tryptic peptides were dissolved in solvent A (0.1% formic acid, 0.1% acetonitrile/in water), directly loaded onto a homemade reversed-phase analytical

column (25 cm length, 75/100 μm i.d.). Peptides were separated with a gradient from 6% to 24% solvent B (0.1% formic acid in acetonitrile) over 70 min, 24% to 35% in 14 min and climbing to 80% in 3 min then holding at 80% for the last 3 min, all at a constant flow rate of 450 nL/min on a nanoElute UHPLC system (Bruker Daltonics). The peptides were subjected to capillary source followed by the timsTOF Pro (Bruker Daltonics) mass spectrometry. The electrospray voltage applied was 1.75 kV. Precursors and fragments were analyzed at the TOF detector, with an MS/MS scan range from 100 to 1,700 m/z. The timsTOF Pro was operated in parallel accumulation serial fragmentation (PASEF) mode. Precursors with charge states 0 to 5 were selected for fragmentation, and 10 PASEF-MS/MS scans were acquired per cycle. The dynamic exclusion was set to 30 s.

Data processing protocol. The resulting MS/MS data were processed using MaxQuant search engine (v1.6.6.0). Tandem mass spectra were searched against the Mus_musculus_10090_SP_20191115 (17,032 entries) concatenated with reverse decoy database. Trypsin/P was specified as cleavage enzyme allowing up to 2 missing cleavages. The mass tolerance for precursor ions was set as 20 ppm in first search and 20 ppm in main search, and the mass tolerance for fragment ions was set as 0.02 Da. Carbamidomethyl on Cys was specified as fixed modification, and acetylation on protein N-terminal and oxidation on Met were specified as variable modifications. FDR was adjusted to <1%.

## Cell aging induction

NIH3T3 cells were cultured in DMEM supplemented with 10% FBS. NIH3T3 cells were trypsinized and suspended in phosphate buffer solution (PBS) with $H_2O_2$ (400 μm in PBS) at 37°C for 45 min. $H_2O_2$ treatment was terminated by 3 times of washing with PBS. Then, the NIH3T3 cells were cultured with complete medium for 3 days.

## Animal experiments

All procedures involve animals were performed in conformity with relevant guidelines and regulations and approved by the Ethics Committee of Sichuan University West China hospital (approval number 20220406005).

## Induction of ALI (acute lung injury)

Male mice aged 8 weeks were randomly divided into 3 groups (control group, LPS (10 mg/kg) group, LPS (10 mg/kg) + LY2874455 (1 mg/kg) group, 6 mice in each group). Mice were subject to intragastric administration with either saline or LY2874455 (1 mg/kg) at the same time as LPS injection (10 mg/kg). Six hours after LPS injection, mice were killed and the lungs were removed. Lung tissues were subsequently used for quantitative real-time PCR assay and hematoxylineosin (HE) staining.

## Induction of sepsis

Male mice aged 8 weeks were randomly divided into 3 groups, 8 mice in each group. To evaluate the effect of LY2874455 on the survival of mice, endotoxemia was induced by a single i.p. injection of bacterial endotoxin (LPS, 35 mg/kg), with i.g. administration of LY2874455 (1 mg/kg), at 2 h before the injection of LPS. The mice not receiving LY2874455 received a mock injection with saline as controls. Survival rates of animals were monitored up to 96 h after LPS challenge.

## Induction of colitis

Colitis was induced in 8-week-old male mice by adding 3% DSS (m.w. 36,000 to 50,000; MP Biomedicals, Solon, Ohio, United States of America) to the drinking water, beginning on day 0. On the third day of DSS treatment, mice were subject to intragastric administration with either saline or LY2874455 (1 mg/kg) for 5 days. Thereafter, mice were given regular drinking water. The body weight was measured daily throughout the experiment. At day 9, the mice were killed and the intestines were removed. Intestine tissues were subsequently used for quantitative real-time PCR assay, western blot assay, and HE staining.

## Poly (IC) model

Polyinosinic-polycytidylic acid (poly(I:C)) (50 μg/ml) and LY2874455 (2 μm) were applied to BMDM for 24 h and expression levels of inflammatory cytokines were analyzed qRT-PCR.

## Transmission electron microscopy

Macrophages were washed 3 times for 15 min with 0.1 M phosphate buffer, and then fixed in 2% aqueous osmium tetraoxide for 1 h followed by washing 3 times each for 15 min with deionized water. Samples were then dyed with 2% uranyl acetate for 30 min and dehydrated through graded alcohols (50% to 100%) and 100% acetone each for 15 min. After that, samples were embedded in EPON 812 resin and cured for 24 h at 37°C, 45°C, 60°C, respectively. Ultrathin (70 nm) sections were obtained by ultrathin slicer machine (Leica) and stained with 2% uranyl acetate and 0.3% lead citrate. Electron microscopy images of the samples were taken using Tecnai G2 Spirit transmission electron microscope (FEI Company). Representative images of at least 3 independent replicated experiments are shown.

## Statistical analysis

All data shown in this study are representative results from at least 3 independent replicated experiments. The data were check by Shapiro–Wilk for analysis of normal distribution. Quantification data are expressed as means ± SD from at least 3 biological replicates. Statistical analyses were conducted using Prism5. Ordinary one-way or two-way ANOVA with Turkey's multiple comparisons test, or a two-tailed paired or unpaired Student's $t$ test, or the Dunn's nonparametric test (for multiple group comparisons) and the Mann–Whitney test (2 group comparisons) were used to determine statistical significance (*: $P < 0.05$, **: $P < 0.01$, ***: $P < 0.001$, NS: no statistical difference).

## Supporting information

**S1 Fig. LY2874455 suppressed inflammation in LPS-stimulated macrophages.** (A) RAW264.7 cells were incubated with 27 chemicals and stimulated with LPS (20 ng/ml) for 24 h. NO concentration in culture supernatant was measured and shown as fold change. (B) The names of the 27 chemicals. (C, D) Different concentrations of LY2874455 were applied to LPS (20 ng/ml)-treated RAW264.7 cells and cell viabilities and levels of NO were measured. (E–G) The expression of proinflammatory genes (IL-6, TNF-α) and iNOS was checked by qRT-PCR treated with LY2874455 (2 μm) and LPS (20 ng/ml) for 24 h. (H) Protein levels of IL-6, TNF-α, iNOS, and β-actin were detected by western blot in RAW264.7 cells. The blot was quantified with densitometric values and statistical significance was analyzed. (I–K) The expression of proinflammatory genes (IL-6, TNF-α) and iNOS was checked by qRT-PCR in PMs treated with LY2874455 (2 μm) and LPS (20 ng/ml) for 24 h. (L) Protein levels of iNOS and GAPDH were detected by western blot in PM. (M) The blot was quantified with densitometric values

and statistical significance was analyzed. (N–P) The expression of proinflammatory genes (IL-6, TNF-α) and iNOS was checked by qRT-PCR in BMDMs treated with LY2874455 (2 μm) and LPS (20 ng/ml) for 24 h. (Q, R) Expression levels of iNOS were detected by in BMDM. The blot was quantified with densitometric values and statistical significance was analyzed. (S, T) The expression of proinflammatory genes (IL-6 and TNF-α) was checked by qRT-PCR in human THP-1 monocytes treated with LY2874455 (2 μm) and LPS (20 ng/ml) for 24 h. Phorbol12-myristate13-acetate (PMA) was used to induce THP-1 monocyte to macrophages. (U, V) The cytokine of IL-6 and TNF-α was detected by elisa in in RAW264.7 cells treated with LY2874455 (2 μm) and LPS (20 ng/ml) for 24 h. (W–Y) The protein expression of IL-6, TNF-α, and iNOS in Fig 1H were quantified with densitometric values and statistical significance was analyzed. (Z–AB) The protein expression of IL-6, TNF-α, and iNOS in S1H Fig were quantified with densitometric values and statistical significance was analyzed. (AC) The protein expression of IKK-β in Fig 1I was quantified with densitometric values and statistical significance was analyzed. (AD–AE) The protein expression of p65 and p-p65 in Fig 1J were quantified with densitometric values and statistical significance was analyzed. (AF) The immune fluorescence micrographs of p65 were quantified in Fig 1K. *: $P < 0.05$, **: $P < 0.01$, ***: $P < 0.001$, ****: $P < 0.0001$, NS: no statistical difference. The data underlying the graphs shown in the figure can be found in S1 Data.
(PDF)

**S2 Fig. LY287445 suppressed viral infection- and aging-induced inflammation.** (A–C) LY2874455 inhibited viral infection-induced inflammation. Polyinosinic-polycytidylic acid (poly(I:C)) (50 μg/ml) and LY2874455 (2 μm) were applied to BMDM for 24 h and expression levels of inflammatory cytokines were analyzed qRT-PCR. (D–F) LY2874455 inhibited the bacteria (*E. coli*) infection-induced inflammation. *E. coli* and LY2874455 (2 μm) were applied to BMDM for 24 h and expression levels of inflammatory cytokines were analyzed qRT-PCR. (G–K) LY2874455 inhibited $H_2O_2$-induced cell aging. NIH3T3 cells were treated with $H_2O_2$ (400 μm for 2 h) for induction of cell aging followed by LY2874455 (2 μm) treatment. Representative images of β-GAL staining of the cells were shown (G) and the expression levels of p16 (H) and inflammatory cytokines (I–K) were analyzed by qRT-PCR. (L) The protein expression of IL1β, IL18, and GAPDH were detected by western blot. And the protein expression was quantified with densitometric values and statistical significance was analyzed. *: $P < 0.05$, **: $P < 0.01$, ***: $P < 0.001$, ****: $P < 0.0001$, NS: no statistical difference. The data underlying the graphs shown in the figure can be found in S1 Data.
(PDF)

**S3 Fig. Immunoproteasome functions in LPS-induced inflammation at upstream of NF-κB factors.** (A) Immunoproteasome inhibitor ONX-0914 suppressed the nucleartranslocation of NF-κB factor p65 in macrophages stimulated with LPS. RAW264.7 cells were stimulated with LPS (20 ng/ml) for 2 h with or without ONX-0914 (1 μm). Representative images of 3 independent replicates were shown. Scale bars: 10 μm. (B) Schematic diagram of NF-κB inflammation signaling pathway. (C) Total proteins were extracted from RAW264.7 macrophage cells treated with LPS (20 ng/ml) together with ONX-0914 (1 μm) and used for western blot analysis of protein levels of phosphorylated IκB-α (p-IκB-α). The blot was quantified with densitometric values and statistical significance was analyzed. (D, E) Total proteins were extracted from RAW264.7 macrophage cells treated with LPS (20 ng/ml) together with ONX-0914 (1 μm) or LY2874455 (2 μm) and used for western blot analysis of protein levels of indicated factors function upstream of the NF-κB signaling pathway. The blots were quantified with densitometric values and statistical significance was analyzed. (F) The enzymatic activities of immunoproteasome subunits LMP10 were measured in RAW264.7 cells. ***: $P < 0.001$.

(G) Total proteins were extracted from RAW264.7 macrophage cells treated with LPS (20 ng/ml) together with LY2874455 (2 μm) and used for western blot analysis of protein levels of PSMB1 and GAPDH. (H) The protein expression of PSMB1 in was quantified with densitometric values and statistical significance was analyzed. (I) The protein expression of LMP2, LMP7, and LMP10 were quantified with densitometric values and statistical significance was analyzed. (J) The protein expression of PSMB6, PSMB7, and PSMB5 in Fig 3E were quantified with densitometric values and statistical significance was analyzed. (K) The protein expression of LMP2, LMP7, and LMP10 in Fig 3F was quantified with densitometric values and statistical significance was analyzed. (L) The protein expression of PSMB6, PSMB7, and PSMB5 in Fig 3G were quantified with densitometric values and statistical significance was analyzed. (M) The immune fluorescence micrographs of p65 in S3A Fig were quantified. (N–S) The protein expression of p-IκB-α, TAK1, MyD88, TAB2, TRIF, and TRAF6 in S3D Fig were quantified with densitometric values and statistical significance was analyzed. (T–X) The protein expression of p-IκB-α, TAK1, MyD88, TAB2, TRIF, and TRAF6 in S3E Fig were quantified with densitometric values and statistical significance was analyzed. *: $P < 0.05$, **: $P < 0.01$, ***: $P < 0.001$, ****: $P < 0.0001$, NS: no statistical difference. The data underlying the graphs shown in the figure can be found in S1 Data.
(PDF)

**S4 Fig. LY2874455 induced autophagy in macrophages.** (A) Inhibitors MG132, chloroquine, or wortmannin were used at different concentrations to analyze their effects on macrophage viabilities in the dose- and time-dependent cell death assays. (B) Inhibitor of constitutive proteasome, MG132, could not block the reduction of immunoproteasome subunits induced by LY2874455. RAW264.7 cells were treated with LPS (20 ng/ml), LY2874455 (2 μm), and proteasome inhibitor MG132 (5 μm). Protein levels of LMP2, LMP7, and GAPDH were detected by western blot. The blots were quantified with densitometric values and statistical significance was analyzed. (C) Autophagy inhibitor wortmannin blocked the reduction of immunoproteasome subunits induced by LY2874455. Protein levels of LMP2, LMP7, and GAPDH were analyzed in RAW264.7 cells treated with LPS (20 ng/ml), LY2874455 (2 μm), and autophagy inhibitor wortmannin (100 nM). The blots were quantified with densitometric values and statistical significance was analyzed. (D) Model scheme of LY2874455 effect on activation of autophagy. FGFRs are transmembrane RTKs and activated by FGF to form dimers, leading to the activation of downstream FRS2 complex and subsequent activation of PI3K/AKT/mTOR signaling pathway. Activated mTOR then phosphorylates and inhibits autophagy initiating ULK1 kinase complex, leading to suppression of autophagy. LY2874455 suppresses the FGFR activities and the downstream PI3K/AKT/mTOR signaling pathway, leading to the release and activation of autophagy. (E) NIH3T3 cells were treated with LPS (20 ng/ml) and LY2874455 (0.2, 0.5, 1, 2, 5 μm) for 24 h and protein levels of p62, LC3, and GAPDH were analyzed by western blot. The blots were quantified with densitometric values and statistical significance was analyzed. (F) AZD4547, an inhibitor of FGFR, was used to detect the effect on autophagy and immunoproteasome. RAW264.7 cells were treated with LPS (20 ng/ml) and ADZ4547 (10, 50, 100 nM) for 24 h and protein levels of p62, LC3, and GAPDH were analyzed by western blot. The blots were quantified with densitometric values and statistical significance was analyzed. (G–I) AZD4547 also has effects on inflammation suppression in RAW264.7 cells. RAW264.7 cells were treated with LPS (20 ng/ml) and ADZ4547 (100 nM) for 24 h, and the expression of IL-6, TNF-α, *IL1β*, and iNOS was checked by qRT-PCR. (J) Chemicals (Fenticonazole, Domiphen, and Cetylpyridinium) were used to detect their effects on autophagy activation. RAW264.7 cells were treated with Fenticonazole, Domiphen, and Cetylpyridinium (0.5, 1, 15 μm) for 24 h and protein levels of p62 and GAPDH were analyzed by western blot.

The blots were quantified with densitometric values and statistical significance was analyzed. (K–M) The protein expression of LMP2 and LMP7 in Fig 4A–4C were quantified with densitometric values and statistical significance was analyzed. (N) The number of LC3 puncta were quantified in Fig 4D. (O) Quantification of autophagosomes in macrophages treated with LY2874455 for results in Fig 4E. (P) The protein expression of p62 and the ratio of LC3II/LC3I in Fig 4F were quantified with densitometric values and statistical significance was analyzed. (Q–S) The protein expression of p62 in Fig 4G–4I were quantified with densitometric values and statistical significance was analyzed. (T–W) The protein expression of p-mTOR (Ser248), mTOR, p-AKT(Ser473), AKT in Fig 4J were quantified with densitometric values and statistical significance was analyzed. (X–AA) The protein expression of LMP2 and LMP7 in S4B and S4C Fig were quantified with densitometric values and statistical significance was analyzed. (AB–AC) The protein expression of p62 and the ratio of LC3II/LC3I in S4E Fig were quantified with densitometric values and statistical significance was analyzed. (AD–AF) The protein expression of LMP2, LMP7, and p62 in S4F Fig were quantified with densitometric values and statistical significance was analyzed. (AG–AI) The protein expression of p62 in S4J Fig was quantified with densitometric values and statistical significance was analyzed. *: $P < 0.05$, **: $P < 0.01$, ***: $P < 0.001$, ****: $P < 0.0001$, NS: no statistical difference. The data underlying the graphs shown in the figure can be found in S1 Data.
(PDF)

**S5 Fig. LY287445 induced the degradation of immunoproteasome subunits through autophagy.** (A) Confirmation of ATG7 gene knock out in macrophages. ATG7 gene was deleted through CRISPR-CAS9 assay in RAW264.7 cells. Clones of gene deletion cells were cultured and detected for protein levels of p62, LC3, ATG7, and intrinsic control GAPDH. The blots were quantified with densitometric values and statistical significance was analyzed. (B) Confirmation of the decrease of p62 protein levels in p62 knock down RAW264.7 cells. The blot was quantified with densitometric values and statistical significance was analyzed. (C) The protein expression of LMP2 and LMP7 in Fig 5A were quantified with densitometric values and statistical significance was analyzed. (D) The protein expression of LMP2, LMP7, and p62 in Fig 5B were quantified with densitometric values and statistical significance was analyzed. (E) The protein expression of LMP2, LMP7, and p62 in Fig 5C were quantified with densitometric values and statistical significance was analyzed. (F) The immune fluorescence micrographs in Fig 5D was quantified by ImageJ. (G) The protein expression of NRB1, Tollip, and p62 in Fig 5E were quantified with densitometric values and statistical significance was analyzed. (H) The protein expression of LMP2 and LMP7 in Fig 5F were quantified with densitometric values and statistical significance was analyzed. *: $P < 0.05$, **: $P < 0.01$, ***: $P < 0.001$, ****: $P < 0.0001$, NS: no statistical difference. The data underlying the graphs shown in the figure can be found in S1 Data.
(PDF)

**S6 Fig. LY287445 suppressed inflammation through autophagy.** (A, B) Autophagy inhibitor wortmannin reversed the effect of LY2874455 in LPS-stimulated macrophages. BMDMs were treated with LPS (20 ng/ml), LY2874455 (2 μm), and wortmannin (100 nM) as indicated for 24 h. The expression of iNOS and inflammatory cytokine genes IL-6 was checked by qRT-PCR. (C) RAW264.7 cells and used for western blot analysis of protein levels of p65, phosphorylated p65 (p-p65), and IKK-β. The blots were quantified with densitometric values and statistical significance was analyzed. (D) RAW264.7 cells were stimulated with LPS (20 ng/ml) for 2 h with or without LY2874455 (2 μm). Representative images of double immunostaining for p65 and nucleus (DAPI) were shown. Scale bars: 10 μm. (D, F) Deficiency of autophagy factor ATG7 or receptor p62 caused blockage of LY2874455 function in suppression of NF-κB

factor p65 (phosphorylation and nucleus translocation). (E, G) Total proteins were extracted from ATG7 knockout or p62 knockdown RAW264.7 cells and used for western blot analysis of protein levels of p65, phosphorylated p65 (p-p65), and IKK-β. The blots were quantified with densitometric values and statistical significance was analyzed. (F, H) ATG7 knockout or p62 knockdown RAW264.7 cells were stimulated with LPS (20 ng/ml) for 2 h with or without LY2874455 (2 μm). Representative images of double immunostaining for p65 and nucleus (DAPI) were shown. Scale bars: 10 μm. (I) The enhanced inflammatory capacity in autophagy-deficient macrophages was suppressed by immunoproteasome inhibitor ONX-0914. Wild-type (WT) or ATG7 knockout (ATG7-KO) RAW264.7 cells treated with LPS (20 ng/ml) and ONX-0914 (1 μm). The protein levels of iNOS and inflammatory cytokine genes (IL-6 and TNF-α) were checked. The blots were quantified with densitometric values and statistical significance was analyzed. (J, K) Autophagy inhibitors (ULK1-IN-2 and chloroquine (CQ)) reversed the effect of LY2874455 in LPS-stimulated macrophages. RAW264.7 cells were harvest after treatment with LPS (20 ng/ml), LY2874455 (2 μm) and ULK1-IN-2 (5 μm) or chloroquine (20 μm) as indicated for 24 h. The expression of inflammatory cytokine genes (iNOS and IL-6) was checked by qRT-PCR in RAW264.7 cells. (L) siRNA was used to knock down the expression of lmp7. The gene expression of lmp7 was detected by qRT-PCR. (M) RAW264.7 cells were transfected with siRNA for 48 h and then treated with ONX-0914 for 24 h. The gene expression of IL6 was detected by qRT-PCR. (N) The protein expression of p65, phosphorylated p65 (p-p65), and IKK-β in S6C Fig were quantified with densitometric values and statistical significance was analyzed. (O) The protein expression of p65, phosphorylated p65 (p-p65), and IKK-β in S6D Fig were quantified with densitometric values and statistical significance was analyzed. (P) The protein expression of p65, phosphorylated p65 (p-p65), and IKK-β in S6G Fig were quantified with densitometric values and statistical significance was analyzed. (Q) The protein expression of iNOS, TNF-α, and IL-6 in S6I Fig were quantified with densitometric values and statistical significance was analyzed. (R) Diagrammatic sketch of selective autophagy of the immunoproteasome suppresses inflammation regulation. *: $P < 0.05$, **: $P < 0.01$, ***: $P < 0.001$, ****: $P < 0.0001$, NS: no statistical difference. The data underlying the graphs shown in the figure can be found in S1 Data.
(PDF)

**S1 Table. Primers and dilutions for antibodies used in this study.**
(PDF)

**S2 Table. Proteins analyzed by mass spectrometry in RAW264.7 macrophages treated with LPS and LY2874455.**
(XLSX)

**S1 Data. Data that underlies all figures.**
(XLSX)

**S1 Raw Images. Uncropped images for all relevant figures in this article.**
(PDF)

# Acknowledgments

We thank the Imaging Center of Sichuan University for imaging.

# Author Contributions

**Conceptualization:** Jiao Zhou, Gaoyue Jiang, Shanze Chen, Huihui Li, Kefeng Lu.

**Data curation:** Chunxia Li.

**Funding acquisition:** Huihui Li, Kefeng Lu.

**Investigation:** Jiao Zhou.

**Methodology:** Jiao Zhou, Chunxia Li, Meng Lu, Gaoyue Jiang.

**Project administration:** Jiao Zhou.

**Resources:** Gaoyue Jiang, Shanze Chen.

**Supervision:** Huihui Li, Kefeng Lu.

**Validation:** Jiao Zhou, Kefeng Lu.

**Writing – original draft:** Jiao Zhou.

**Writing – review & editing:** Kefeng Lu.

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
