## [Editor Report · Decision Letter 0]

19 Feb 2023

Dear Dr Zhou, 

Thank you for submitting your manuscript entitled "Selective Autophagy of the Immunoproteasome Suppresses Innate Inflammatory Signalling" for consideration as a Research Article by PLOS Biology.

Your manuscript has now been evaluated by the PLOS Biology editorial staff as well as by an academic editor with relevant expertise and I am writing to let you know that we would like to send your submission out for external peer review.

Once your full submission is complete, your paper will undergo a series of checks in preparation for peer review. After your manuscript has passed the checks it will be sent out for review. To provide the metadata for your submission, please Login to Editorial Manager (https://www.editorialmanager.com/pbiology) within two working days, i.e. by Feb 21 2023 11:59PM.

Kind regards,

Ines

--

Ines Alvarez-Garcia, PhD

Senior Editor

PLOS Biology

---

## [Decision Letter · Decision Letter 1]

27 Apr 2023

Dear Dr Zhou,

Thank you for your patience while your manuscript entitled "Selective Autophagy of the Immunoproteasome Suppresses Innate Inflammatory Signalling" was peer-reviewed at PLOS Biology. Also please accept my sincere apologies for the delay in providing you with our decision. Your manuscript has been evaluated by the PLOS Biology editors, an Academic Editor with relevant expertise, and by three independent reviewers.

The reviews are attached below. As you will see, the reviewers find the work potentially interesting, but they have also raised several important concerns that would have to be address in order for us to consider the manuscript for publication. The issues include the specificity of the FGFR and immunoproteasome inhibitors and the fact that prior related literature has not been discussed, which would be essential in a revision.

Based on the specific reviewers' comments and following discussion with the Academic Editor, it is clear that a substantial amount of work would be required to meet the criteria for publication in PLOS Biology. However, given the interest, we would be open to inviting a comprehensive revision of the study that thoroughly addresses all the reviewers' comments. As the extent of revision that would be needed is considerable, we cannot make a decision about publication until we have seen the revised manuscript and your response to the reviewers' comments. Your revised manuscript would need to be seen by the reviewers again, but please note that we would not engage them unless their main concerns have been addressed. 

We appreciate that these requests represent a great deal of extra work, and we are willing to relax our standard revision time to allow you 6 months to revise your study. Please email us (plosbiology@plos.org) if you have any questions or concerns, or envision needing a (short) extension.

**IMPORTANT - SUBMITTING YOUR REVISION**

3. Resubmission Checklist

a) *PLOS Data Policy*

b) *Published Peer Review*

d) *Blurb*

Please also provide a blurb which (if accepted) will be included in our weekly and monthly Electronic Table of Contents, sent out to readers of PLOS Biology, and may be used to promote your article in social media. The blurb should be about 30-40 words long and is subject to editorial changes. It should, without exaggeration, entice people to read your manuscript. It should not be redundant with the title and should not contain acronyms or abbreviations. For examples, view our author guidelines: https://journals.plos.org/plosbiology/s/revising-your-manuscript#loc-blurb

Sincerely,

Ines

--

Ines Alvarez-Garcia, PhD

Senior Editor

PLOS Biology

Reviewers' comments

Rev. 1:

The manuscript by Zhou et al. presents a detailed analysis and interesting findings on the FGFR inhibitor activated autophagy inhibiting inflammation through degradation of immunoproteasomes in macrophages. All the data provided are based on different experimental approaches using macrophage cells and animal models. The experiments are well designed and are clearly presented in this manuscript. The authors first screened and found FGFR inhibitor inhibits inflammation and promotes specific degradation of immunoproteasome in macrophages, then the authors found this inhibitor efficiently activates autophagy, which mediates the effects of this inhibitor in inflammation suppression and immunoproteasome degradation, furthermore, the authors uncovered ubiquitination as the trigger and p62 as the receptor for this selective autophagy of immunoproteasome. With two mouse models, the authors found this inhibitor can function for inflammation suppression in vivo. All in all, the presented data are well documented and very interesting, linking the selective autophagy of immunoproteasome with inflammation suppression in macrophages. I believe that this study could pave the way for new investigations into the connection between autophagy and innate inflammatory signaling and could, therefore, be considered to be published in PLOS Biology after minor revision. Specific points that help improve this manuscript are described below.

1, The FGFR inhibitor LY2874455 is interesting, can the authors try to detect more other inhibitors of FGFR, to see whether such effects on inflammation suppression, autophagy of immunoproteasome are still conferred by other inhibitors of FGFR? This is important since this will confirm the effects of autophagy of immunoproteasome in inflammation suppression is not only conferred by one inhibitor.

2, The authors focused on macrophages in this study, and data provided are strong enough to support the conclusions. I am curious that whether this FGFR inhibitor can also exert function of autophagy activation in other types of cells, such as fibroblasts?

3, In Fig.1, the authors found FGFR inhibitor, also several other chemicals were found to suppress inflammation such as #8, 10, 19, with less efficiency (Fig.S1A). Can the authors analyze these chemicals to detect their effects on autophagy activation？

4，In Fig.2, the authors analyzed subunits of constituent proteasome, PSMB1, 2, 5, can the authors analyze other subunits? As these three subunits are counterparts of LAMP2, 7, 10, by analyzing other subunits, it will be more convinced that it is the whole constituent proteasome being degraded at the presence of FGFR inhibitor LY2874455.

5, In Fig.3F, the p62 blots showed two bands at high levels, is that simply because of the high amount of p62? It is suggested to be repeated this.

6, The imaging pictures should be organized more carefully, for example, in Fig. 1K and Fig. 2D, the panel pictures are not aligned. Similarly, the WB blots should also be organized more carefully, for example, in Fig. S5G, the upper two panels should be reorganized. I suggest the authors go through all the data to check the format.

Rev. 2: Christian Münz – note that this reviewer has signed his review

Zhou et al., "Selective Autophagy of the Immunoproteasome Suppresses Innate Inflammatory Signalling"

The authors identified (R)-(E)-2-(4-(2-(5-(1-(3,5-dichloropyridin-4-yl) ethoxy)-1H180 indazol-3yl) vinyl)-1H-pyrazol-1-yl) ethanol (LY2874455) as a compound that limits nitric oxide (NO), reactive oxygen species (ROS), TNF and IL-6 production by the mouse macrophage cell line RAW264.7 upon LPS stimulation. A similar phenotype has been observed in mouse peritoneal and bone marrow derived macrophages and the human monocyte cell line THP-1. This compound has previously been reported to be a pan-fibroblast growth factor receptor (FGFR) inhibitor. They demonstrate that LY2874455 stabilizes IKK-beta preventing NF-kappaB translocation to the nucleus and pro-inflammatory mediator production after LPS stimulation. The proteomic analysis of LY2874455 treated RAW264.7 cells revealed depletion of immunoproteasomes and their inhibition with ONX-0914 phenocopies the hypoinflammation upon FGFR inhibition. Furthermore, the authors determine with wortmannin and chloroquine inhibition that immunoproteasomes are turned over by autophagy, and that LY2874455 increases autophagic flux of the mCherry-EGFP-LC3 reporter construct and the p62 autophagy receptor. ATG7 deficiency and RNA silencing of p62 increased immunoproteasome levels. Finally, the authors demonstrate that LY287445 reduces LPS induced lung and DSS induced bowel inflammation, and in supplemental data on polyI:C and age induced inflammmation. Therefore, the authors suggest that FGFR inhibition induces autophagy that degrades immunoproteasomes which in turn are required for NF-kappaB dependent inflammation.

The study is interesting and provides many mechanistic insights into the role of autophagy and immunoproteasomes in inflammation. However, it suffers from not discussing prior literature on autophagic mechanisms that limit inflammation and proteasomal degradation by autophagy, both of which diminish the novelty of the reported findings. Furthermore, the specificity of the FGFR and immunoproteasome inhibitors are not controlled for by respective knock-outs and knock-downs and only surrogate pathogen associated molecular pattern (PAMP) molecules are used to mimic bacterial and viral infections.

Major comments:

1. The authors introduce inflammatory tissue damage during COVID-19 and sepsis but then use primarily LPS challenge. It would be interesting if immunoproteasome degradation would influence inflammation upon exposure to bacteria or viruses. Along these lines the authors should also not call polyI:C induced inflammation viral infection and H2O2 exposure age-related inflammation, in the text referring to supplemental figure 6.

2. The presented data with the two pharmacological inhibitors for FGFR and immunoproteasomes should be confirmed at least for RAW264.7 produced inflammatory mediators by RNA silencing or knock-out for FGFR or immunoproteasome subunits. Even so LY2874455 and ONX-0914 might be very specific for their targets only the absence of their effects upon FGFR or immunoproteasome deficiency, respectively, would clearly document this.

3. Inflammasome regulation by autophagy has been studied by many investigators as mechanism of hyperinflammation upon autophagy deficiency in myeloid cells. Therefore, the authors should assess inflammasome output, such as IL-1beta and IL-18, upon LY2874455 treatment and its likely independence of ONX-0914 mediated immunoproteasome inhibition. This would also confirm specificity of the used inhibitors.

4. Some of the immune fluorescence micrographs should be quantified and stated how many experiments were performed, such as the nuclear p65 translocation in Figures 1K, S2A, S5D and S5F, the autophagy reporter in Figure 3D, and co-localization of immunoproteasome subunits with LC3 in Figure 4D.

5. The authors identify FGFR inhibition as stimulus to increase autophagic flux. As they discuss, autophagy stimulation would be beneficial in many disease settings and, therefore, well tolerated autophagy inducers are of great interest. However, previous literature has suggested that FGF signaling actually induces autophagy in some cell types, such as chondrocytes (Cinque et al., Nature 2015). Therefore, any insights that the authors can provide on why FGFR inhibition causes autophagy up-regulation in their experimental system, while FGFR signaling stimulates autophagy in other cell types would clarify a broad application of FGFR inhibitors to stimulate autophagy.

Minor comments:

1. Some typos: line 127, achieved instead of chieved; line 463, Beclin1 instead of Belin1

2. There is quite some literature on mechanisms by which autophagy restricts inflammation, regulating inflammasomes, STING and MAVS for cytosolic pathogen associated molecular pattern (PAMP) recognition. Therefore, the statement in lines 140 to 141 should be corrected and these previously described mechanisms acknowledged.

3. Similarly no previous literature on autophagy of proteasomes is cited. This should be done, e.g. Sengupta et al., EMBO J 2019 and Dengjel et al., Mol Cell Proteomics 2012.

Rev. 3: Anna Katharina Simon - note that this reviewer has signed her review

General comments

The authors demonstrate that the FGFR inhibitor LY2874455 can limit production of pro-inflammatory cytokines by LPS-stimulated macrophages through selective degradation of immunoproteasome subunits via autophagy. This finding is novel and provides an additional explanation for exaggerated inflammatory processes observed upon autophagy-deficiency. The experiments in this study are well designed and executed. Especially, the use of primary murine macrophages and bone-marrow derived macrophages in addition to the RAW264.7 cell line and confirmation of the efficacy of the inhibitor in different in vivo models are major assets of the study. Besides grammar and spelling of the manuscript that have to be improved (some of them have been indicated in the pdf document) and inaccuracies in the method section, we would like to see the following major and minor points addressed in the revised manuscript.

Major comments

Point 1

The authors state, that treatment with LY2874455 did not further repress production of inflammatory cytokines in ONX-0914-treated macrophages in Figure 2I and J (Lines 248-250). However, the graph and the figure legend indicate, that cells were either treated with LY2874455 or ONX-0914, thus making this statement invalid. Furthermore, a combination of these treatments would not prove, that LY2874455 acts through inhibition of the immunoproteasome, as ONX-0914 inhibition already abolishes production of pro-inflammatory cytokines, which cannot thus not be further reduced by LY2874455. We ask the authors to a) correct the description of the aforementioned figure and b) comment on their conclusion.

Point 2

We are missing a statement on normal distribution of the data in the statistical analysis paragraph (starting in line 744). This concerns all data in the manuscript. Which tests were used to probe for normal distribution of the data? In some graphs, normal distribution of the data is questionable and thus statistical analysis could be incorrect.

Point 3

Throughout the figure legends, it was mentioned that western blot signals were quantified with densitometric values and statistical significance was analyzed. However, while the ratios of densitometric values of protein of interest and housekeeping protein are provided in the graphs, the results of the statistical analysis are missing. It is also not clear how often the western blot experiments were reproduced with independent replicates. Therefore, we ask the authors to add the quantification and statistical analysis for all western blots displayed and indicate how many replicates were analyzed for all western blots displayed throughout the manuscript.

Point 4

The method used in Figure 3D is not well explained in the results (line 275-279) or method section or in the figure legend. We assume, that the GFP signal is quenched upon autophagosome-lysosome fusion, but the mCherry signal is not affected. However, this is not clearly stated by the authors and the explanation given in the figure legend is quite confusing. Also, statistical analysis of this figure is missing and the results are questionable as no lysosomal inhibitor was used, which would reveal if there was indeed stalling of the autophagic process or rather increased LC3 expression. Therefore, we consider the conclusion drawn in lines 286-288 (stating that LPS blocked autophagic flux) invalid. A western blot detecting LC3I and II and the use of a lysosomal fusion inhibitor such as Bafilomycin A would be the appropriate control experiment.

Point 5

The methods proving that autophagy is indeed activated upon treatment with LY2874455 as displayed in figure 3 are not completely sufficient. A) Some of the arrows in figure 3E are not pointing at autophagosomes, but rather lysosomes. Therefore, the quantification presented in Figure S3D is questionable. B) Quantification of LC3I and II in addition to p62 would be needed to actually prove induction of involvement of autophagy.

Finally, wortmannin is not considered the best agent for autophagy inhibition, as it targets other processes. The use of chloroquine, as presented in figure 3B, is preferred and we therefore wonder, why Chloroquine was not used in the experiments presented in Figure 5. Also, use of a more specific autophagy inhibitor such as ULK-1 inhibitor would be a better option.

Point 6

We have serious doubts about the validity of only using GAPDH (an enzyme in the glycolysis pathway) as a housekeeping gene and/or protein for all qRT-PCR and western blot experiments. It has been demonstrated by various studies, that GAPDH expression is triggered by deletion or inhibition of autophagy, LPS stimulation of macrophages and other treatment of cells performed in this manuscript. Thus, we ask the authors if they have used additional housekeeping proteins / transcripts as a control and if the results obtained using other control proteins / transcripts were similar.

The results could also benefit from additional analysis of cytokine concentrations in the supernatant using ELISA, as this is considered more reliable than mRNA expression, especially for cytokines.

Point 7

In our opinion, the DSS-induced colitis model is not properly evaluated. A) We wondered whether the authors did not measure mRNA expression or protein expression of IL-6 and IL-1b. B) To properly evaluate this model, other parameters should be provided like a histology scoring of the intestine, as well as changes in body weight of the mice.

Point 8

We would like to suggest to restructure the manuscript in a way, that the in vivo data is mentioned before mechanism.

Point 9

We feel, that the title is overinterpreting the results, as there are not many convincing experiments using autophagy KO or inhibition alone, and the autophagy inducing compound used may affect more processes than autophagy. As an alternative title we suggest: Pharmacological induction of autophagy reduces inflammation in macrophages by degrading immunoproteasome subunits.

Minor comments

Point 1

The writing of the manuscript has to be improved. There are a lot of spelling mistakes and grammar errors, which complicate the reading of the manuscript. Some of them are addressed in the commented version of the PDF.

Point 2

Throughout the manuscript, the authors refer to the observation made using macrophage stimulation in vitro to changes in "inflammation". However, this cannot be stated as only expression of pro-inflammatory factors (mostly on mRNA level) was investigated and inflammation rather refers to the biological process in the body. We ask the authors to rephrase the respective sentences.

Point 3

The Introduction contains a lot of information which is not needed for understanding the manuscript. Especially the part from line 65 to 106, where regulation of RTKs is explained in detail could be omitted, and instead papers could be cited more in detail that actually deal with immunoproteasome and inflammation.

Point 4

Several references are missing throughout the manuscript. Those are indicated in the commented version of the PDF.

Point 5

The authors state, that 5618 components were tested for potential inhibition of NO production by LPS-stimulated RAW264.7 macrophages. However, only the results of 27 are shown and there is no comment on the results of the other compounds. We are missing a statement here, whether NO production was unaffected by all other components. Additionally, it would be nice to provide the results for all components tested as a supplementary file to the scientific community.

Point 6

Line 270: The results of figures S3A and B are not mentioned in the text.

Point 7

The last three lanes of Figure 3F are not explained in the text (see lines 284-286).

Point 8

The authors state in line 297, that mTOR inhibits autophagy by initiating ULK1 kinase complex. However, this is not true. mTOR targets Ulk1 but inhibits the activation of the complex through inhibitory phosphorylation, as it is correctly stated in the figure legend of Fig. S3. This should be corrected.

Point 9

In Figure 4, most of the blots do not contain annotation for molecular weights. Please provide those.

Point 10

To us, it is not clear how figure 4G proves ubiquitination of LMP2 and LMP7, as there are no bands around 50 kDa in the right upper panel of Figure 4G. We ask the authors to comment on that.

Point 11

Please split the axis of Figure S5B to visualize the difference between LPS and LPS+LY2874455.

Point 12

The poly (IC) model used in Figure S6A-C is not mentioned or explained in the methods section. Additionally, this is not an actual model of viral infection but rather a model mimicking viral infection. Please add description of the model and amend the statement in line 398.

Point 13

The statement in lines 400-402 was not investigated and thus not confirmed in the mouse models mentioned in this paragraph. Therefore, this statement should be deleted or amended.

Point 14

The statement on immunoproteasome-induced inflammatory processes provided in lines 428 to 430 seems oversimplified and is missing a reference. Please add more specific information and cite studies proving this connection.

Point 15

Materials and Reagents paragraph (starting line 523): If it is allowed by the journal guidelines, a table would be much more comprehensive instead of the text written in this paragraph.

Point 16

It is unclear if the statement given on injection of thioglycolate in line 552 means, that the injections took place 3 days prior to the sacrifice, or 3 days in a row followed by the sacrifice of the mice.

Point 17

There are several unclear or missing statements on procedures, composition of media or manufacturers in the methods sections which are marked in the commented version of the PDF.

Point 18

We wondered whether approval numbers provided by authorities have to be mentioned in the animal experiments section (line 695-698).

Point 19

From the description in the methods section (lines 709-715) it is unclear, whether the mice not receiving LY2874455 received a mock injection with e.g. saline as a control. Please add a sentence clarifying this issue.

Point 20

A display of wildtype cells in figure S5C-F is missing, which is in our opinion needed to prove that the experiment was successful and p65 does translocate to the nucleus upon LPS treatment in presence of autophagy.

---

## [Decision Letter · Decision Letter 2]

23 Dec 2023

Dear Dr Zhou,

Thank you for your patience while we considered your revised manuscript entitled "Pharmacological induction of autophagy reduces inflammation in macrophages by degrading immunoproteasome subunits" for publication as a Research Article at PLOS Biology. This revised version of your manuscript has been evaluated by the PLOS Biology editors, the Academic Editor and two of the original reviewers.

Based on the reviews, we are likely to accept this manuscript for publication, provided you satisfactorily address the remaining points raised by Reviewer 2. Please also make sure to address the following data and other policy-related requests stated below.

We expect to receive your revised manuscript within two weeks. 

*Published Peer Review History*

*Press*

Sincerely,

Ines

--

Ines Alvarez-Garcia, PhD

Senior Editor

PLOS Biology

Fig. 1B, D, E-G; Fig. 2B, C, E, F, H-L; Fig. 3A, H-J; Fig. 6A-J; Fig. S1A, C-G, I-K, M-P, R-AF; Fig. S2A-F, H-L; Fig. S3F, H-W, S; Fig. S4A, G-AI; Fig. S5C-H and Fig. S6A, B, D, F, H, J-Q

Please also ensure that figure legends in your manuscript include information ON WHERE THE UNDERLYING DATA CAN BE FOUND, and ensure your supplemental data file/s has a legend.

We require the original, uncropped and minimally adjusted images supporting all blot and gel results reported in an article's figures or Supporting Information files. We will require these files before a manuscript can be accepted so please prepare and upload them now. Please carefully read our guidelines for how to prepare and upload this data: https://journals.plos.org/plosbiology/s/figures#loc-blot-and-gel-reporting-requirements

DATA NOT SHOWN

Reviewers' comments

Rev. 1:

The authors addressed all my questions.

Rev. 2:

The authors identified (R)-(E)-2-(4-(2-(5-(1-(3,5-dichloropyridin-4-yl) ethoxy)-1H180 indazol-3yl) vinyl)-1H-pyrazol-1-yl) ethanol (LY2874455) as a compound that limits nitric oxide (NO), reactive oxygen species (ROS), TNF and IL-6 production by the mouse macrophage cell line RAW264.7 upon LPS stimulation. A similar phenotype has been observed in mouse peritoneal and bone marrow derived macrophages and the human monocyte cell line THP-1. This compound has previously been reported to be a pan-fibroblast growth factor receptor (FGFR) inhibitor. They demonstrate that LY2874455 stabilizes IKK-beta preventing NF-kappaB translocation to the nucleus and pro-inflammatory mediator production after LPS stimulation. The proteomic analysis of LY2874455 treated RAW264.7 cells revealed depletion of immunoproteasomes and their inhibition with ONX-0914 phenocopies the hypoinflammation upon FGFR inhibition. Furthermore, the authors determine with wortmannin and chloroquine inhibition that immunoproteasomes are turned over by autophagy, and that LY2874455 increases autophagic flux of the mCherry-EGFP-LC3 reporter construct and the p62 autophagy receptor. ATG7 deficiency and RNA silencing of p62 increased immunoproteasome levels. Finally, the authors demonstrate that LY287445 reduces LPS induced lung and DSS induced bowel inflammation, and in supplemental data on polyI:C and age induced inflammmation. Therefore, the authors suggest that FGFR inhibition induces autophagy that degrades immunoproteasomes which in turn are required for NF-kappaB dependent inflammation.

In their revised manuscript, the authors have addressed most of my major concerns, namely used E. coli exposure in addition to LPS and toned down the interpretation of polyI:C and H2O2. Furthermore, they performed specificity controls for their inhibitors and observe also reduced inflammasome activity in addition to immunoproteasome dependent NF-kappaB activity upon autophagy stimulation. They have now also quantified the immune fluorescence microscopy images.

The authors have also addressed two of my three minor concerns but somehow have not answered to my minor point #2 in the point by point. Therefore, I would like them to acknowledge the previous literature on autophagic suppression of inflammation and the respective mechanisms to clarify that their mechanism is one of many and that their study is not the first that documents hypoinflammation upon autophagy stimulation.

Minor concern:

1. There is quite some literature on mechanisms by which autophagy restricts inflammation, regulating inflammasomes, STING and MAVS for cytosolic pathogen associated molecular pattern (PAMP) recognition. Therefore, the statement in lines 401 to 402 should be corrected and these previously described mechanisms acknowledged.

---

## [Editor Report · Decision Letter 3]

5 Feb 2024

Dear Dr Zhou,

Thank you for the submission of your revised Research Article entitled "Pharmacological induction of autophagy reduces inflammation in macrophages by degrading immunoproteasome subunits" for publication in PLOS Biology. On behalf of my colleagues and the Academic Editor, Hans-Uwe Simon, I am pleased to say that we can in principle accept your manuscript for publication, provided you address any remaining formatting and reporting issues. These will be detailed in an email you should receive within 2-3 business days from our colleagues in the journal operations team; no action is required from you until then. Please note that we will not be able to formally accept your manuscript and schedule it for publication until you have completed any requested changes.

PRESS

Sincerely, 

Ines

--

Ines Alvarez-Garcia, PhD

Senior Editor

PLOS Biology
